# CHARACTERIZING LOOKAHEAD DYNAMICS OF SMOOTH GAMES

## ABSTRACT

As multi-agent systems proliferate in machine learning research, games have attracted much attention as a framework to understand optimization of multiple interacting objectives. However, a key challenge in game optimization is that, in general, there is no guarantee for usual gradient-based methods to converge to a local solution of the game. The latest work by Chavdarova et al. (2020) report that Lookahead optimizer (Zhang et al., 2019) significantly improves the performance of Generative Adversarial Networks (GANs) and reduces the rotational force of bilinear games. While promising, their observations were purely empirical, and Lookahead optimization of smooth games still lacks theoretical understanding. In this paper, we fill this gap by theoretically characterizing Lookahead dynamics of smooth games. We provide an intuitive geometric explanation on how and when Lookahead can improve game dynamics in terms of stability and convergence. Furthermore, we present sufficient conditions under which Lookahead optimization of bilinear games provably stabilizes or accelerates convergence to a Nash equilibrium of the game. Finally, we show that Lookahead optimizer preserves locally asymptotically stable equilibria of base dynamics, and can either stabilize or accelerate the local convergence to a given equilibrium with proper assumptions. We verify our theoretical predictions by conducting numerical experiments on two-player zero-sum (non-linear) games.

## 1 INTRODUCTION

Recently, a plethora of learning problems have been formulated as games between multiple interacting agents, including Generative Adversarial Networks (GANs) (Goodfellow et al., 2014; Brock et al., 2019; Karras et al., 2019), adversarial training (Goodfellow et al., 2015; Madry et al., 2018), self-play (Silver et al., 2018; Bansal et al., 2018), inverse reinforcement learning (RL) (Fu et al., 2018) and multi-agent RL (Lanctot et al., 2017; Vinyals et al., 2019). However, the optimization of interdependent objectives is a non-trivial problem, in terms of both computational complexity (Daskalakis et al., 2006; Chen et al., 2009) and convergence to an equilibrium (Goodfellow, 2017; Mertikopoulos et al., 2018; Mescheder et al., 2018; Hsieh et al., 2020). In particular, gradient-based optimization methods often fail to converge and oscillate around a (local) Nash equilibrium of the game even in a very simple setting (Mescheder et al., 2018; Daskalakis et al., 2018; Mertikopoulos et al., 2019; Gidel et al., 2019b;a). To tackle such non-convergent game dynamics, a huge effort has been devoted to developing efficient optimization methods with nice convergence guarantees in smooth games (Mescheder et al., 2017; 2018; Daskalakis et al., 2018; Balduzzi et al., 2018; Gidel et al., 2019b;a; Schäfer & Anandkumar, 2019; Yazici et al., 2019; Loizou et al., 2020).

Meanwhile, Chavdarova et al. (2020) have recently reported that the Lookahead optimizer (Zhang et al., 2019) significantly improves the empirical performance of GANs and reduces the rotational force of a bilinear game dynamics. Specifically, they demonstrate that class-unconditional GANs trained by a Lookahead optimizer can outperform class-conditional BigGAN (Brock et al., 2019) trained by Adam (Kingma & Ba, 2015) even with a model of $1/30$ parameters and negligible computation overheads. They also show that Lookahead optimization of a stochastic bilinear game tends to be more robust against large gradient variances than other popular first-order methods, and converges to a Nash equilibrium of the game where other methods fail.

Despite its great promise, the study of Chavdarova et al. (2020) relied on purely empirical observations, and the dynamics of Lookahead game optimization still lacks theoretical understanding. Specifically, many open questions, such as the convergence properties of Lookahead dynamics and the impact of its hyperparameters on the convergence, remain unexplained. In this work, we fill this gap by theoretically characterizing the Lookahead dynamics of smooth games. Our contributions are summarized as follows:

- We provide an intuitive geometric explanation on how and when Lookahead can improve the game dynamics in terms of stability and convergence to an equilibrium.
- We analyze the convergence of Lookahead dynamics in bilinear games and present sufficient conditions under which the base dynamics can be either stabilized or accelerated.
- We characterize the limit points of Lookahead dynamics in terms of their stability and local convergence rates. Specifically, we show that Lookahead (i) preserves locally asymptotically stable equilibria of base dynamics and (ii) can either stabilize or accelerate the local convergence to a given equilibrium by carefully choosing its hyperparameters.
- Each of our theoretical predictions is verified with numerical experiments on two-player zero-sum (non-linear) smooth games.

## 2 PRELIMINARIES

We briefly review the objective of smooth game optimization, first-order game dynamics, and Lookahead optimizer. Finally, we discuss previous work on game optimization. We summarize the notations throughout this paper in Table A.1.

### 2.1 SMOOTH GAMES

Following Balduzzi et al. (2018), a smooth game between players $i = 1, \ldots, n$ can be defined as a set of smooth scalar functions $\{f_i\}_{i=1}^n$ with $f_i : \mathbb{R}^d \to \mathbb{R}$ such that $d = \sum_{i=1}^n d_i$. Each $f_i$ represents the cost of player $i$'s strategy $\mathbf{x}_i \in \mathbb{R}^{d_i}$ with respect to other players' strategies $\mathbf{x}_{-i}$. The goal of this game optimization is finding a (local) Nash equilibrium of the game (Nash, 1951), which is a strategy profile where no player has an unilateral incentive to change its own strategy.

**Definition 1** (Nash equilibrium). *Let $\{f_i\}_{i=1}^n$ be a smooth game with strategy spaces $\{\mathbb{R}^{d_i}\}_{i=1}^n$ such that $d = \sum_{i=1}^n d_i$. Then $\boldsymbol{x}^* \in \mathbb{R}^d$ is a local Nash equilibrium of the game if, for each $i = 1, \ldots, n$, there is a neighborhood $U_i$ of $\boldsymbol{x}_i^*$ such that $f_i(\boldsymbol{x}_i, \boldsymbol{x}_{-i}^*) \geq f_i(\boldsymbol{x}^*)$ holds for any $\boldsymbol{x}_i \in U_i$. Such $\boldsymbol{x}^*$ is said to be a global Nash equilibrium of the game when $U_i = \mathbb{R}^{d_i}$ for each $i = 1, \ldots, n$.*

A straightforward computational approach to find a (local) Nash equilibrium of a smooth game is to carefully design a gradient-based strategy update rule for each player. Such update rules that define iterative *plays* between players are referred to as a *dynamics* of the game.

**Definition 2** (Dynamics of a game). *A dynamics of a smooth game $\{f_i\}_{i=1}^n$ indicates a differentiable operator $F : \mathbb{R}^d \to \mathbb{R}^d$ that describes players' iterative strategy updates as $\boldsymbol{x}^{(t+1)} = F(\boldsymbol{x}^{(t)})$.*

One might expect that a simple myopic game dynamics, such as gradient descent, would suffice to find a (local) Nash equilibrium of a game as in traditional minimization problems. However, in general, gradient descent optimization of smooth games often fail to converge and oscillate around an equilibrium of the game (Daskalakis et al., 2018; Gidel et al., 2019b;a; Letcher et al., 2019). Such non-convergent behavior of game dynamics is mainly due to (non-cooperative) interaction between multiple cost functions, and is considered as a key challenge in the game optimization (Mescheder et al., 2017; 2018; Mazumdar et al., 2019; Hsieh et al., 2020).

### 2.2 FIRST-ORDER METHODS FOR SMOOTH GAME OPTIMIZATION

We introduce well-known first-order methods for smooth game optimization. To ease the notation, we use $\nabla_{\mathbf{x}}\mathbf{f}(\cdot)$ to denote the concatenated partial derivatives $(\nabla_{\mathbf{x}_1} f_1(\cdot), \ldots, \nabla_{\mathbf{x}_n} f_n(\cdot))$ of a smooth game $\{f_i\}_{i=1}^n$, where $\nabla_{\mathbf{x}_i} f_i(\cdot)$ is a partial derivative of a player $i$'s cost function with respective to its own strategy.

**Gradient Descent (GD)** minimizes the cost function of each player using the gradient descent. Its simultaneous dynamics $F_{\text{GD}_{\text{Sim}}}$ of a smooth game $\{f_i\}_{i=1}^n$ with a learning rate $\eta > 0$ is given by

$$\mathbf{x}^{(t+1)} = F_{\text{GD}_{\text{Sim}}}(\mathbf{x}^{(t)}) \stackrel{\text{def}}{=} \mathbf{x}^{(t)} - \eta \nabla_{\mathbf{x}} \mathbf{f}(\mathbf{x}^{(t)}). \tag{1}$$

On the other hand, its alternating dynamics $F_{\text{GD}_{\text{Alt}}}$ is described by

$$\mathbf{x}^{(t+1)} = F_{\text{GD}_{\text{Alt}}}(\mathbf{x}^{(t)}) \stackrel{\text{def}}{=} F_1 \circ \ldots \circ F_n(\mathbf{x}^{(t)}), \quad \text{where} \tag{2}$$

$$F_i(\mathbf{x}) \stackrel{\text{def}}{=} (\ldots, x_{i-1}, x_i - \eta \nabla_{x_i} f_i(\mathbf{x}), x_{i+1}, \ldots). \tag{3}$$

**Proximal Point (PP)** (Martinet, 1970) computes an update by solving a proximal problem at each iteration. Its simultaneous dynamics $F_{\text{PP}_{\text{Sim}}}$ of a smooth game $\{f_i\}_{i=1}^n$ with a learning rate $\eta > 0$ is

$$\mathbf{x}^{(t+1)} = F_{\text{PP}_{\text{Sim}}}(\mathbf{x}^{(t)}) \stackrel{\text{def}}{=} \mathbf{x}^{(t)} - \eta \nabla_{\mathbf{x}} \mathbf{f}(\mathbf{x}^{(t+1)}). \tag{4}$$

Note that this update rule is implicit in a sense that $\mathbf{x}^{(t+1)}$ appears on both sides of the equation; hence it requires solving the proximal subproblem for $\mathbf{x}^{(t+1)}$ per iteration.

**Extra Gradient (EG)** (Korpelevich, 1976) computes an update by using an *extrapolated* gradient. Its simultaneous dynamics $F_{\text{EG}_{\text{Sim}}}$ of a smooth game $\{f_i\}_{i=1}^n$ with a learning rate $\eta > 0$ is

$$\mathbf{x}^{(t+1)} = F_{\text{EG}_{\text{Sim}}}(\mathbf{x}^{(t)}) \stackrel{\text{def}}{=} \mathbf{x}^{(t)} - \eta \nabla_{\mathbf{x}} \mathbf{f}(\mathbf{x}^{(t+\frac{1}{2})}), \quad \text{where} \tag{5}$$

$$\mathbf{x}^{(t+\frac{1}{2})} \stackrel{\text{def}}{=} \mathbf{x}^{(t)} - \eta \nabla_{\mathbf{x}} \mathbf{f}(\mathbf{x}^{(t)}). \tag{6}$$

## 2.3 LOOKAHEAD OPTIMIZER

Lookahead (Zhang et al., 2019) is a recently proposed optimizer that wraps around a base optimizer and takes a *backward* synchronization step for each $k$ *forward* steps. Given a dynamics $F_{\mathcal{A}}$ induced by a base optimization method $\mathcal{A}$, the Lookahead dynamics $G_{\text{LA-}\mathcal{A}}$ with a synchronization period $k \in \mathbb{N}$ and a rate $\alpha \in (0, 1)$ is

$$\mathbf{x}^{(t+1)} = G_{\text{LA-}\mathcal{A}}(\mathbf{x}^{(t)}) \stackrel{\text{def}}{=} (1 - \alpha)\mathbf{x}^{(t)} + \alpha F_{\mathcal{A}}^k(\mathbf{x}^{(t)}). \tag{7}$$

## 2.4 RELATED WORK

The convergence analysis of first-order smooth game dynamics dates several decades back and have been established in the context of saddle-point problems (Rockafellar, 1976; Korpelevich, 1976; Tseng, 1995), which is a special case of zero-sum games. For example, Rockafellar (1976) showed the linear convergence of PP in the bilinear and strongly-convex-strongly-concave (SCSC) saddle-point problems. Tseng (1995) and Facchinei & Pang (2003) proved the linear convergence of EG in the same problem, and Nemirovski (2004) did in the convex-concave problem over compact sets.

As many learning problems are formulated as games in recent years (Goodfellow et al., 2014; Madry et al., 2018; Silver et al., 2018; Fu et al., 2018; Vinyals et al., 2019), game optimization has regained considerable attentions from the research community. Optimistic gradient descent (OGD) (Popov, 1980), which can be seen as an efficient approximation of EG, was recently rediscovered in the context of GAN training (Daskalakis et al., 2018). Recent work of Liang & Stokes (2019) and Gidel et al. (2019a) proved linear convergence of OGD in bilinear and SCSC games. Mokhtari et al. (2020) established an unifying theoretical framework for analyzing PP, EG and OGD dynamics. Zhang & Yu (2020) presented exact and optimal conditions for PP, EG and OGD dynamics to converge in bilinear games. While there has been a growing interest for incorporating second-order information into game dynamics (Mescheder et al., 2017; Balduzzi et al., 2018; Mazumdar et al., 2019; Schäfer & Anandkumar, 2019; Loizou et al., 2020) to remedy non-convergent behaviors, the first-order optimization still dominates in practice (Brock et al., 2019; Donahue & Simonyan, 2019) due to computational and memory cost of second-order methods.

Lately, Chavdarova et al. (2020) reported that recently developed Lookahead optimizer (Zhang et al., 2019) significantly improves the empirical performance of GANs and reduces the rotational force of bilinear game dynamics. However, this study relied on purely empirical observation and lacked theoretical understanding for Lookahead optimization of smooth games. Although Wang et al. (2020) proved that Lookahead optimizer globally converges to a stationary point in minimization problems, its convergence in smooth games still remain as an open question.

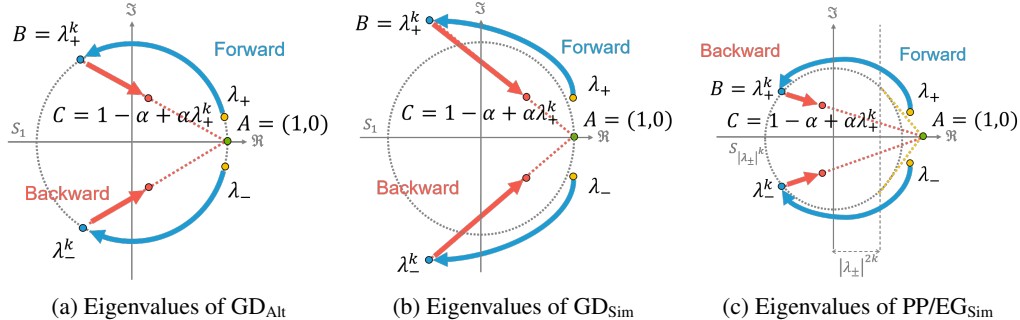

(a) Eigenvalues of GD$_{\text{Alt}}$    (b) Eigenvalues of GD$_{\text{Sim}}$    (c) Eigenvalues of PP/EG$_{\text{Sim}}$

Figure 1: The *spectral contraction effect* of Lookahead dynamics in Equation 8. $\lambda_{\pm}$ are the eigenvalues of each base dynamics, and $1 - \alpha + \alpha\lambda_{\pm}^k$ are the eigenvalues of the associated Lookahead dynamics. $k$ *forward* steps of a Lookahead procedure first rotate the eigenvalues $\lambda_{\pm}$ of the dynamics' Jacobian matrix. Then, a synchronization *backward* step pulls them into a circle with a radius smaller than their maximal modulus. This results in a reduced spectral radius of the Jacobian matrix, which improves stability and convergence to an equilibrium.

## 3 Spectral Contraction Effect of Lookahead in Bilinear Games

In this section, we show that Lookahead can either stabilize or accelerate the convergence of its base dynamics by reducing the spectral radius of its underlying Jacobian matrix. We highlight such *spectral contraction effect* by analyzing the convergence of Lookahead dynamics in a simple bilinear game (Section 3.1), and extend the results to general bilinear games (Section 3.2).

### 3.1 Lookahead Dynamics of a Simple Bilinear Game

We begin with a simple exemplar bilinear game that has a unique Nash equilibrium $(0, 0)$:

$$\min_{x_1 \in \mathbb{R}} \max_{x_2 \in \mathbb{R}} \; x_1 \cdot x_2. \tag{8}$$

This game has been extensively studied as a representative toy example of game optimization by Gidel et al. (2019a) due to its oscillating dynamics. The following proposition demonstrates stabilization effect of Lookahead on Equation 8.

**Proposition 1.** *Simultaneous GD dynamics $F_{GD_{Sim}}$ with a learning rate $\eta > 0$ diverges from the Nash equilibrium of Equation 8. However, its Lookahead dynamics $G_{LA\text{-}GD_{Sim}}$ with a synchronization period $k \in \mathbb{N}$ and a rate $\alpha \in (0, 1)$ globally converges to the Nash equilibrium if $\Re((1 + i\eta)^k) < 1$ and $\alpha$ is small enough.*

Proposition 1 shows that Lookahead optimizer can stabilize divergent dynamics of Equation 8. However, such stabilization effect of Lookahead raises a natural question: would there be any advantage for using Lookahead when its base dynamics is already stable? Proposition 2 analyzes well-known convergent PP dynamics of Equation 8 and presents an affirmative answer. Specifically, it shows that Lookahead dynamics (i) preserves the convergence of its base dynamics, and (ii) can further accelerate the convergence with proper hyperparameter choices.

**Proposition 2.** *Simultaneous PP Lookahead dynamics $G_{LA\text{-}PP_{Sim}}$ with a learning rate $\eta \in (0, 1)$, a synchronization period $k \in \mathbb{N}$ and a rate $\alpha \in (0, 1)$ globally converges to the Nash equilibrium of Equation 8. Furthermore, the rate of convergence is improved upon its base dynamics $F_{PP_{Sim}}$ if $\Re((1 + i\eta)^k) < (1 + \eta^2)^k$ and $\alpha$ is large enough.*

We provide geometric interpretation of the Lookahead procedure in Figure 1. Intuitively, Lookahead optimizer either stabilizes or accelerates its base dynamics by pulling the eigenvalues of the dynamics' Jacobian matrix into a circle with a small radius. Specifically, $k$ *forward* steps of a Lookahead procedure first rotate the eigenvalues, and a synchronization *backward* step pulls them into a circle with a radius smaller than their maximal modulus. This results in a reduction of the spectral radius of the dynamics' Jacobian matrix, which is known to be crucial for stability (Slotine & Li, 1991). Such *spectral contraction effect* of Lookahead dynamics is captured by the following lemma.

**Lemma 3** (Spectral contraction effect of Lookahead). *Let $k \in \mathbb{N}$, $\alpha \in (0, 1)$ and define a function $f : \mathbb{R}^{m \times m} \to \mathbb{R}^{m \times m}$ by $f(\boldsymbol{X}) = (1 - \alpha)\boldsymbol{I} + \alpha\boldsymbol{X}^k$. Define $\theta(\lambda) \stackrel{\text{def}}{=} \mathrm{Arg}(\lambda^k - 1)$ and $\phi(\lambda) \stackrel{\text{def}}{=} \arcsin \frac{\sin(\theta(\lambda))}{\rho(\boldsymbol{X})^k}$. Then, the following statements hold:*

- *For $\rho(\boldsymbol{X}) = 1$, $\rho(f(\boldsymbol{X})) < 1$ if $\lambda^k \neq 1$, $\forall \lambda \in \lambda_{max}(\boldsymbol{X})$.*

- *For $\rho(\boldsymbol{X}) > 1$, $\rho(f(\boldsymbol{X})) < 1$ if $\Re(\lambda^k) < 1$, $\alpha < \frac{2\cos(\pi - \theta(\lambda))}{|\lambda^k - 1|}$, $\forall \lambda \in \lambda_{\geq 1}(\boldsymbol{X})$.*

- *For $\rho(\boldsymbol{X}) < 1$, $\rho(f(\boldsymbol{X})) < \rho(\boldsymbol{X})^k$ if $\Re(\lambda^k) < \rho(\boldsymbol{X})^{2k}$, $\alpha > 1 - \frac{2\rho(\boldsymbol{X})^k \cos(\pi - \phi(\lambda_i))}{|\lambda_i^k - 1|}$, $\forall \lambda \in \lambda_{max}(\boldsymbol{X})$, $\forall \lambda_i \in \lambda(\boldsymbol{X})$.*

In short, Lemma 3 suggests that Lookahead can reduce the spectral radius of a matrix by choosing a proper $\alpha$ and a $k$ such that the entire *radius-supporting eigenvalues (e.g., $\lambda_{\geq 1}(\mathbf{X})$, $\lambda_{\max}(\mathbf{X})$) are rotated to left enough.* However, such $k$ may not exist, for example, especially when such eigenvalues are not tightly clustered together. To help understanding when Lookahead can actually reduce the spectral radius, we present Lemma 4 as a sufficient condition for a set of eigenvalues to admit the existence of $k$ that rotates them to the left half-plane.

**Lemma 4** (Left-rotatable eigenvalues). *Let $X, J \in \mathbb{R}^{m \times m}$ be such that $X = I - \eta J$ for some $\eta > 0$ and let $S \subseteq \lambda(X)$. Assume that each element of $S$ has its conjugate pair in $S$. Then we have $\Re(\lambda^k) < 0$ for each $\lambda \in S$ if $k \in \left( \frac{\pi}{2\theta_{min}(S)}, \frac{3\pi}{2\theta_{max}(S)} \right)$ and every element of $S$ has non-zero imaginary part. Existence of such $k \in \mathbb{N}$ is guaranteed for a small enough $\eta$ when $\frac{\Im_{max}(S)}{\Im_{min}(S)} < 3$.*

Note that the Jacobian matrix of most well-known gradient-based dynamics can be written in the form of $\mathbf{I} - \eta\mathbf{J}$, where $\eta > 0$ is a learning rate and $\mathbf{J}$ is the underlying Jacobian matrix of the game. Intuitively, Lemma 4 suggests that for a small enough learning rate, any *subset* of the eigenvalues of a dynamics with imaginary conditioning less than 3 admits the existence of $k$ that rotates them *left enough.* For such $k$, Lookahead can reduce the spectral radius of the dynamics by choosing a proper $\alpha$, as stated in Lemma 3. This joint usage of Lemma 3-4 plays a central role for the proofs of our main results in Section 3.2 and Section 4. To summarize, Lemma 3-4 together highlight when Lookahead can actually improve the game dynamics and show that the imaginary conditioning of the *radius-supporting eigenvalues* is crucial for determining whether the dynamics is *improvable*.

## 3.2 Lookahead Dynamics of General Bilinear Games

In this section, we extend the analysis of Lookahead dynamics to a general bilinear game

$$\min_{\mathbf{x}_1 \in \mathbb{R}^m} \max_{\mathbf{x}_2 \in \mathbb{R}^n} \mathbf{x}_1^T \mathbf{A} \mathbf{x}_2 - \mathbf{b}_1^T \mathbf{x}_1 - \mathbf{b}_2^T \mathbf{x}_2 \qquad (9)$$

for some $\mathbf{A} \in \mathbb{R}^{m \times n}$ and $\mathbf{b}_1 \in \mathbb{R}^m, \mathbf{b}_2 \in \mathbb{R}^n$ such that there exists $\mathbf{x}_1^* \in \mathbb{R}^m, \mathbf{x}_2^* \in \mathbb{R}^n$ with $\mathbf{A}^T \mathbf{x}_1^* = \mathbf{b}_2$ and $\mathbf{A}\mathbf{x}_2^* = \mathbf{b}_1$. The existence of $\mathbf{x}_1^*, \mathbf{x}_2^*$ allows us to rewrite the game as

$$\min_{\mathbf{x}_1 \in \mathbb{R}^m} \max_{\mathbf{x}_2 \in \mathbb{R}^n} (\mathbf{x}_1 - \mathbf{x}_1^*)^T \mathbf{U} \begin{bmatrix} \Sigma_r & \mathbf{0} \\ \mathbf{0} & \mathbf{0} \end{bmatrix} \mathbf{V}^T (\mathbf{x}_2 - \mathbf{x}_2^*), \qquad (10)$$

where $\mathbf{U}, \Sigma_r, \mathbf{V}$ is the SVD of $\mathbf{A}$ with $r \stackrel{\text{def}}{=} \mathrm{rank}(\mathbf{A})$. Therefore, we can analyze the dynamics of Equation 9 by inspecting a rather simpler problem

$$\min_{\mathbf{x}_1 \in \mathbb{R}^r} \max_{\mathbf{x}_2 \in \mathbb{R}^r} \mathbf{x}_1^T \Sigma_r \mathbf{x}_2, \qquad (11)$$

as they are equivalent up to some rotations and translations. This reduction is a well-known technique and has been used by Gidel et al. (2019b;a) and Zhang & Yu (2020) for simplifying the analysis of Equation 9.

Now we present sufficient conditions for Lookahead hyperparameters under which convergence of each first-order base dynamics, namely GD$_{\text{Alt}}$, GD$_{\text{Sim}}$, PP$_{\text{Sim}}$ and EG$_{\text{Sim}}$, is either stabilized or accelerated. The following first two theorems show that Lookahead can provably stabilize non-convergent GD dynamics of general bilinear games.

**Theorem 5** (Convergence of $G_{\text{LA-GD}_{\text{Alt}}}$). *Lookahead dynamics $G_{\text{LA-GD}_{\text{Alt}}}$ with a learning rate $\eta \in \left(0, \frac{2}{\sigma_{max}}\right)$, a synchronization period $k \in \mathbb{N}$ and a rate $\alpha \in (0, 1)$ converges to a Nash equilibrium of Equation 9 if $k \arccos(1 - \frac{\eta^2 \sigma_i^2}{2}) \mod 2\pi \neq 0$ for any $\sigma_i \in \sigma(\boldsymbol{A})$.*

**Theorem 6** (Convergence of $G_{\text{LA-GD}_{\text{Sim}}}$). *Lookahead dynamics $G_{\text{LA-GD}_{\text{Sim}}}$ with a learning rate $\eta > 0$, a synchronization period $k \in \mathbb{N}$ and a rate $\alpha \in (0, 1)$ converges to a Nash equilibrium of Equation 9 if $k \in \left(\frac{\pi}{2 \arctan \eta \sigma_{min}}, \frac{3\pi}{2 \arctan \eta \sigma_{max}}\right)$ and $\alpha$ is small enough.*

Roughly, Theorem 5 suggests that almost any configurations of Lookahead can make GD$_{\text{Alt}}$ convergent to a Nash equilibrium of the bilinear games. On the other hand, the existence of $k$ that satisfies the condition of Theorem 6 is guaranteed for a small enough $\eta$ if $\frac{\sigma_{\max}}{\sigma_{\min}} < 3$ holds. This highlights a limitation of the convergence guarantee for GD$_{\text{Sim}}$ that it holds only for well-conditioned games.

The next two theorems show that Lookahead preserves the convergence of PP$_{\text{Sim}}$ and EG$_{\text{Sim}}$ in the bilinear games, and can further accelerate their convergence under proper hyperparameter choices.

**Theorem 7** (Acceleration of $G_{\text{LA-PP}_{\text{Sim}}}$). *Lookahead dynamics $G_{\text{LA-PP}_{\text{Sim}}}$ with a learning rate $\eta > 0$, a synchronization period $k \in \mathbb{N}$ and a rate $\alpha \in (0, 1)$ converges to a Nash equilibrium of Equation 9. Furthermore, the rate of convergence is accelerated upon its base dynamics $F_{PP_{Sim}}$ if $k \in \left(\frac{\pi}{2 \arctan \eta \sigma_{min}}, \frac{3\pi}{2 \arctan \eta \sigma_{min}}\right)$ and $\alpha$ is large enough.*

**Theorem 8** (Acceleration of $G_{\text{LA-EG}_{\text{Sim}}}$). *Lookahead dynamics $G_{\text{LA-EG}_{\text{Sim}}}$ with a learning rate $\eta \in \left(0, \frac{1}{\sigma_{max}}\right)$, a synchronization period $k \in \mathbb{N}$ and a rate $\alpha \in (0, 1)$ converges to a Nash equilibrium of Equation 9. Furthermore, the rate of convergence is accelerated upon its base dynamics $F_{EG_{Sim}}$ if $\eta \in \left(0, \frac{1}{2\sigma_{max}}\right), k \in \left(\frac{\pi}{2 \arctan \frac{\eta \sigma_{min}}{1 - \eta \sigma_{min}}}, \frac{3\pi}{2 \arctan \frac{\eta \sigma_{min}}{1 - \eta \sigma_{min}}}\right)$ and $\alpha$ is large enough.*

Note that the existence of $k$ that satisfies the acceleration conditions of Theorem 7-8 is always guaranteed for a small enough $\eta$. This contrasts Theorem 7-8 with Theorem 6, which only applies to well-conditioned games, and suggests that they can be applied for a wide range of bilinear games, including the ill-conditioned ones.

## 4 THE LIMIT POINTS OF LOOKAHEAD DYNAMICS

In this section, we characterize the limit points of Lookahead dynamics and reveal the connections between their stability and the hyperparameters of Lookahead. We start by defining a few stability concepts which are standard in the dynamical system theory (Slotine & Li, 1991).

**Definition 3** (Lyapunov stability). *Let $F$ be a smooth vector field on $\mathbb{R}^n$. Then $\boldsymbol{x} \in \mathbb{R}^n$ is Lyapunov stable in $F$ if for any $\epsilon > 0$, there exists $\delta > 0$ such that for any $\boldsymbol{y} \in \mathbb{R}^n$, $\|\boldsymbol{x} - \boldsymbol{y}\| < \delta$ implies $\|F^t(\boldsymbol{x}) - F^t(\boldsymbol{y})\| < \epsilon$ for all $t \in \mathbb{N}$.*

**Definition 4** (Asymptotic stability). *A Lyapunov stable equilibrium $\boldsymbol{x}^* \in \mathbb{R}^n$ of a smooth vector field $F$ is said to be asymptotically stable if there exists $\delta > 0$ such that $\|\boldsymbol{x} - \boldsymbol{x}^*\| < \delta$ implies $\lim_{t \to \infty} \|F^t(\boldsymbol{x}) - \boldsymbol{x}^*\| = 0$. Such $\boldsymbol{x}^*$ is said to be locally asymptotically stable if $\delta < \infty$.*

We show that any Lyapunov stable equilibrium (SE) of a dynamics is a locally asymptotically stable equilibrium (LASE) of a Lookahead dynamics. Furthermore, we show that Lookahead can either stabilize or accelerate the local convergence to an equilibrium when the *radius-supporting eigenvalues* of the equilibrium satisfy certain assumptions on their imaginary parts.

**Theorem 9** ($\text{SE}_{\mathcal{A}} \subseteq \text{LASE}_{\text{LA-}\mathcal{A}}$). *Let $\boldsymbol{x}^* \in \mathbb{R}^n$ be a Lyapunov stable equilibrium of a dynamics $F$. Then, $\boldsymbol{x}^*$ is a LASE of its Lookahead dynamics $G$ with a synchronization period $k \in \mathbb{N}$ and a rate $\alpha \in (0, 1)$ if $\lambda_i^k \neq 1$ for each $\lambda_i \in \lambda(\nabla_{\boldsymbol{x}} F(\boldsymbol{x}^*))$.*

**Theorem 10** (One-point local stabilization). *Let $\boldsymbol{x}^* \in \mathbb{R}^n$ be an equilibrium of a dynamics $F$ with $\rho(\nabla_{\boldsymbol{x}} F(\boldsymbol{x}^*)) > 1$. Assume that every element of $\lambda_{\geq 1}(\nabla_{\boldsymbol{x}} F(\boldsymbol{x}^*))$ has non-zero imaginary part. Then, $\boldsymbol{x}^*$ is a LASE of its Lookahead dynamics $G$ with a synchronization period $k \in \mathbb{N}$ and a rate $\alpha \in (0, 1)$ if $k \in \left(\frac{\pi}{2\theta_{min}(\lambda_{\geq 1}(\nabla_{\boldsymbol{x}} F(\boldsymbol{x}^*)))}, \frac{3\pi}{2\theta_{max}(\lambda_{\geq 1}(\nabla_{\boldsymbol{x}} F(\boldsymbol{x}^*)))}\right)$ and $\alpha$ is small enough.*

**Theorem 11** (One-point local acceleration). *Let $\boldsymbol{x}^* \in \mathbb{R}^n$ be an equilibrium of a dynamics $F$ with $\rho(\nabla_{\boldsymbol{x}} F(\boldsymbol{x}^*)) < 1$. Assume that every element of $\lambda_{max}(\nabla_{\boldsymbol{x}} F(\boldsymbol{x}^*))$ has non-zero imaginary part. Then, the local convergence rate to $\boldsymbol{x}^*$ in a Lookahead dynamics $G$ with a synchronization period $k \in \mathbb{N}$ and a rate $\alpha \in (0, 1)$ is accelerated upon $F$ if $k \in \left( \frac{\pi}{2\theta_{min}(\lambda_{max}(\nabla_{\boldsymbol{x}} F(\boldsymbol{x}^*)))}, \frac{3\pi}{2\theta_{max}(\lambda_{max}(\nabla_{\boldsymbol{x}} F(\boldsymbol{x}^*)))} \right)$ and $\alpha$ is large enough.*

Intuitively, Theorem 9 shows that Lookahead preserves stability of its base dynamics, and Theorem 10-11 suggest that Lookahead can either stabilize or accelerate the local convergence to an equilibrium. Note that the stabilization and acceleration can be guaranteed when $\lambda_{\geq 1}(\nabla_{\mathbf{x}} F(\mathbf{x}^*))$ and $\lambda_{\max}(\nabla_{\mathbf{x}} F(\mathbf{x}^*))$ contain no real eigenvalues and have imaginary conditioning less than 3; otherwise, $k$ that satisfies the conditions of Theorem 10-11 may not exist (see Appendix E.10-E.11).

An additional, but important consequence of Theorem 10 is that the inclusion relationship implied by Theorem 9 is strict in general. In the context of Nash equilibrium (NE) computation, such stabilization effect of Lookahead can be helpful when unstable NE are stabilized (*e.g.*, bilinear games). However, the stabilization effect also carries a possibility for introducing non-Nash LASE, which is bad for the NE computation (Mazumdar et al., 2019). Hence, the overall impact of Theorem 10 on the computation of NE depends on the global structure of the game and base dynamics.

Note that Theorem 10-11 require radius-supporting eigenvalues to have non-zero imaginary parts and therefore does not apply to *fully-cooperative* (FC) games (*i.e.*, minimization problems), which exhibit real eigenvalues only. To give an understanding of Lookahead dynamics in FC games, we present Proposition 12-13, together which imply that the iterates of Lookahead dynamics almost surely avoids unstable equilibria of its base dynamics in FC games (*e.g.*, avoids local maxima).

**Proposition 12** (Avoids unstable points). *Let $F$ be a $L$-Lipschitz smooth dynamics for some $L > 0$ and let $G$ be its Lookahead dynamics with a synchronization period $k \in \mathbb{N}$ and a rate $\alpha \in \left(0, \frac{1}{1+L^k}\right)$. Then the random-initialized iterates of $G$ almost surely avoids its equilibrium $\boldsymbol{x}^*$ with $\rho(\nabla_{\boldsymbol{x}} G(\boldsymbol{x}^*)) > 1$ if $\rho(\nabla_{\boldsymbol{x}} G(\boldsymbol{x}_0)) \neq 1$ holds for any equilibrium $\boldsymbol{x}_0$ of $G$.*

**Proposition 13** (Preserves unstable points in FC games). *Let $\boldsymbol{x}^* \in \mathbb{R}^n$ be an equilibrium of a dynamics $F$ with $\rho(\nabla_{\boldsymbol{x}} F(\boldsymbol{x}^*)) > 1$, and assume that $\nabla_{\boldsymbol{x}} F(\boldsymbol{x}^*)$ is a symmetric matrix with positive eigenvalues. Then, $\rho(\nabla_{\boldsymbol{x}} G(\boldsymbol{x}^*)) > 1$ holds for a Lookahead dynamics $G$ with a synchronization period $k \in \mathbb{N}$ and a rate $\alpha \in (0, 1)$.*

## 5 EXPERIMENTS

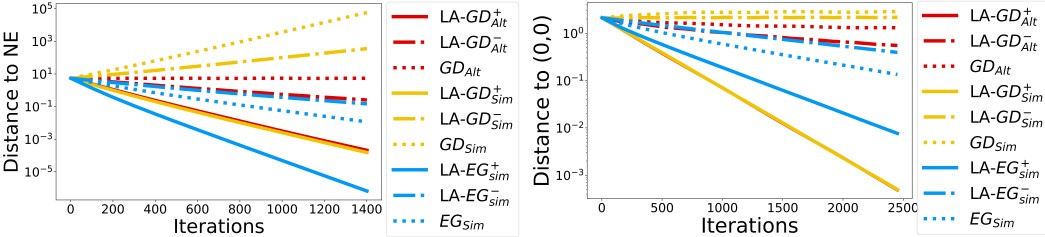

(a) Convergence to the NE of the bilinear game.  (b) Convergence to (0, 0) in the nonlinear game.

Figure 2: Optimization progress of multiple first-order methods with hyperparameters chosen by $(+)$ and against our theorems $(-)$ in the bilinear and nonlinear games.

**Bilinear game**. We test our theoretical predictions in Section 3.2 (Theorem 5–8) on a bilinear game

$$\min_{\mathbf{x}_1 \in \mathbb{R}^n} \max_{\mathbf{x}_2 \in \mathbb{R}^n} \mathbf{x}_1^T \mathbf{A} \mathbf{x}_2 \tag{12}$$

with $\mathbf{A} \stackrel{\text{def}}{=} \mathbf{I}_n + \epsilon \cdot \mathbf{E}_n$, where each element of $\mathbf{E}_n \in \mathbb{M}_{n \times n}$ is sampled from $\mathcal{N}(0, 1)$. We report our results using $n = 10$ and $\epsilon = 0.05$, which gives a sample of $\mathbf{A}$ with $\sigma_{\max} = 1.195$ and $\sigma_{\min} = 0.852$, hence $\frac{\sigma_{\max}}{\sigma_{\min}} = 1.401 < 3$. For a fixed $\eta = 0.1$, we use Theorem 5–8 to derive a range of $k$

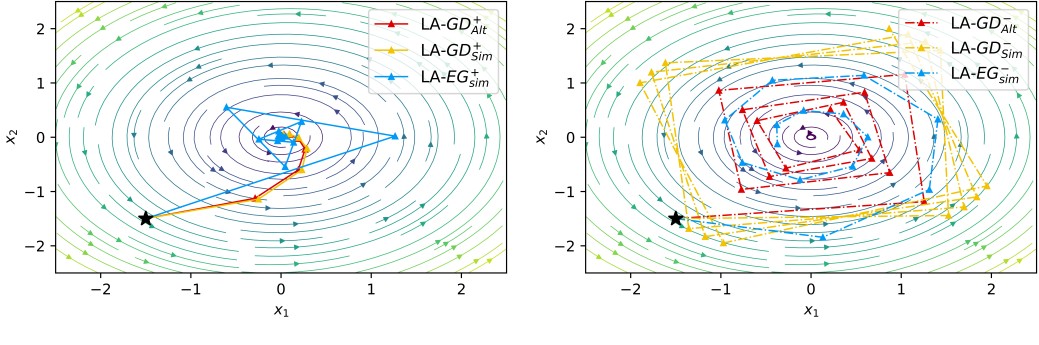

(a) Trajectories of each Lookahead dynamics with $k$ and $\alpha$ predicted by Theorem 10 and 11.

(b) Trajectories of each Lookahead dynamics with $k$ and $\alpha$ chosen against Theorem 10 and 11.

Figure 3: Visualized trajectories of each dynamics an equilibrium $(0,0)$ of the nonlinear game.

and an approximate scale of $\alpha$ that guarantee stabilization and acceleration of convergence to a Nash equilibrium (NE) of Equation 12. We provide the derivations of theoretically recommended values and actual configurations used for the experiment in Appendix D. Figure 2 (a) shows that the hyperparameters predicted by our theorems, denoted by LA-GD$_{\text{Alt/Sim}}^+$ and LA-EG$_{\text{Sim}}^+$, actually stabilize and accelerates the convergence to a NE. We also test the hyperparameters that are chosen against our theorems and denote as LA-GD$_{\text{Alt/Sim}}^-$ and LA-EG$_{\text{Sim}}^-$. Specifically, we choose a $k$ smaller than the lower bound predicted by our theorems and use large $\alpha$ for unstable base dynamics and small $\alpha$ for stable base dynamics. The result in Figure 2 (a) suggests that Lookahead can fail to stabilize, or even worse, slow down the convergence when hyperparameters are configured badly.

**Nonlinear game**. We verify our theoretical predictions in Section 4 (Theorem 10 and 11) on the non-linear game proposed by Hsieh et al. (2020):

$$\min_{x_1 \in \mathbb{R}} \max_{x_2 \in \mathbb{R}} \; x_1 \cdot x_2 + \epsilon\phi(x_2), \tag{13}$$

where $\phi(x) \overset{\text{def}}{=} \frac{1}{2}x^2 - \frac{1}{4}x^4$ with $\epsilon > 0$. This game has an unstable critical point $(0,0)$ surrounded by an attractive internally chain-transitive (ICT) set, which may contain arbitrarily long trajectories. Hsieh et al. (2020) demonstrate that most first-order methods fail to converge in this game due to the instability of the equilibrium and the existence of the ICT set. For a fixed $\epsilon = 0.01$ and $\eta = 0.05$, we use Theorem 10 and 11 to derive a range of $k$ and an approximate scale of $\alpha$ that guarantee local stabilization and acceleration to the equilibrium of Equation 13. We provide the detailed derivations of the theoretically recommended values and the configurations in Appendix D. Figure 2 (b) and Figure 3 (a) shows that the hyperparameters predicted by our theorems, denoted by LA-GD$_{\text{Alt/Sim}}^+$ and LA-EG$_{\text{Sim}}^+$, actually stabilize and accelerates the convergence to the equilibrium. In contrast, hyperparameters chosen against our theorems, denoted by LA-GD$_{\text{Alt/Sim}}^-$ and LA-EG$_{\text{Sim}}^-$ in Figure 2 (b) and Figure 3 (b), neither success to stabilize nor accelerate the convergence to the equilibrium.

## 6 CONCLUSION

In this work, we derived the theoretic results for convergence guarantee and acceleration of Lookahead dynamics in smooth games for the first time. Specifically, we derived sufficient conditions for hyperparameters of Lookahead optimizer under which the convergence of bilinear games is either stabilized or accelerated. Furthermore, we proved that the Lookahead optimizer preserves locally asymptotically stable equilibria of smooth games. Finally, we showed that Lookahead can either stabilize or accelerate the local convergence to a given equilibrium under proper assumptions.

Our results point to several future research directions. Lemma 4 suggests that the imaginary conditioning of the radius-supporting eigenvalues is crucial for the performance gain in Lookahead. Therefore, developing an optimizer that exhibits a small imaginary conditioning could improve the convergence of its Lookahead dynamics. Another interesting application of our theoretic results would be designing an adaptive mechanism for the Lookahead hyperparameters by applying our theorems on local bilinear approximation (Schäfer & Anandkumar, 2019) of the game for each step.

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

## A  NOTATION

Table A.1: List of mathematical notations used in the paper.

| Symbol | Definition |
|---|---|
| $\|\cdot\|$ | $L^2$ norm |
| $x_i$ | $i$-th element of a vector $\mathbf{x} = (x_1, \ldots, x_n)$ |
| $\mathbf{x}_i$ | $i$-th vector of a concatenated vectors $\mathbf{x} = (\mathbf{x}_1, \ldots, \mathbf{x}_n)$ |
| $\mathbf{x}_{-i}$ | $(\mathbf{x}_1, \ldots, \mathbf{x}_{i-1}, \mathbf{x}_{i+1}, \ldots, \mathbf{x}_n)$ |
| $\nabla_x f(x')$ | Derivative of a function $f$ evaluated at $x'$ |
| $S_r$ | The zero-centered circle of radius $r > 0$ in $\mathbb{C}$ |
| $\Re(z)$ | Real part of $z \in \mathbb{C}$ |
| $\Im(z)$ | Imaginary part of $z \in \mathbb{C}$ |
| $\text{Arg}(z)$ | The angle between $z \in \mathbb{C}$ and real axis of the complex plane |
| $\sigma(\mathbf{A})$ | The set of singular values of $\mathbf{A} \in \mathbb{R}^{m \times n}$ |
| $\rho(\mathbf{A})$ | The spectral radius of $\mathbf{A} \in \mathbb{R}^{m \times m}$ |
| $\lambda(\mathbf{A})$ | The set of eigenvalues of $\mathbf{A} \in \mathbb{R}^{m \times m}$ |
| $\lambda_{\geq a}(\mathbf{A})$ | The set of eigenvalues of $\mathbf{A}$ with modulus larger than or equal to $a \in \mathbb{R}$ |
| $\lambda_{\max}(\mathbf{A})$ | The set of eigenvalues of $\mathbf{A}$ with the largest modulus |
| $\Re(S)$ | $\{\Re(c)|c \in S\}$ for $S \subseteq \mathbb{C}$ |
| $\Re_{\min}(S)$ | $\min \Re(S)$ |
| $\Re_{\max}(S)$ | $\max \Re(S)$ |
| $\Im(S)$ | $\{\Im(c)|c \in S\}$ for $S \subseteq \mathbb{C}$ |
| $\Im_{\geq 0}(S)$ | $\{\Im_i \in \Im(S)|\Im_i \geq 0\}$ |
| $\Im_{\min}(S)$ | $\min \Im_{\geq 0}(S)$ |
| $\Im_{\max}(S)$ | $\max \Im_{\geq 0}(S)$ |
| $\theta(S)$ | $\{\text{Arg}(c)|c \in S)\}$ for $S \subseteq \mathbb{C}$ |
| $\theta_{\geq 0}(S)$ | $\{\theta_i|\theta_i \in \theta(S), \theta_i \geq 0\}$ |
| $\theta_{\min}(S)$ | $\min \theta_{\geq 0}(S)$ |
| $\theta_{\max}(S)$ | $\max \theta_{\geq 0}(S)$ |

## B  USEFUL FACTS

### B.1  STANDARD RESULTS ON CONVERGENCE

**Lemma 14** (Bertsekas (1999)). *Let $F : \mathbb{R}^m \to \mathbb{R}^m$ be continuously differentiable, and let $\boldsymbol{x}^* \in \mathbb{R}^m$ be such that $F(\boldsymbol{x}^*) = \boldsymbol{x}^*$. Assume that $\rho(\nabla_{\boldsymbol{x}} F(\boldsymbol{x}^*)) < 1$. Then, there is an open neighborhood $U_{\boldsymbol{x}^*}$ of $\boldsymbol{x}^*$ such that for any $\boldsymbol{x} \in U_{\boldsymbol{x}^*}$, $\|F^t(\boldsymbol{x}) - \boldsymbol{x}^*\|_2 \in \mathcal{O}(\rho(\nabla_{\boldsymbol{x}} F(\boldsymbol{x}^*))^t)$ for $t \to \infty$.*

**Lemma 15** (Gidel et al. (2019b)). *Let $\boldsymbol{M} \in \mathbb{R}^{m \times m}$ and $\boldsymbol{u}^{(t)}$ be a sequence of iterates such that, $\boldsymbol{u}^{(t+1)} = \boldsymbol{M}\boldsymbol{u}^{(t)}$, then we have three cases of interest for the spectral radius $\rho(\boldsymbol{M})$:*

- *If $\rho(\boldsymbol{M}) < 1$ and $\boldsymbol{M}$ is diagonalizable [1], then $\|\boldsymbol{u}^{(t)}\|_2 \in \mathcal{O}(\rho(\boldsymbol{M})^t \|\boldsymbol{u}^{(0)}\|_2)$.*

- *If $\rho(\boldsymbol{M}) > 1$, then there exists $\boldsymbol{u}^{(0)}$ such that $\|\boldsymbol{u}^{(t)}\|_2 \in \Omega(\rho(\boldsymbol{M})^t \|\boldsymbol{u}^{(0)}\|_2)$.*

- *If $|\lambda_i| = 1, \forall \lambda_i \in \lambda(\boldsymbol{M})$, and $\boldsymbol{M}$ is diagonalizable, then $\|\boldsymbol{u}^{(t)}\|_2 \in \Theta(\|\boldsymbol{u}^{(0)}\|_2)$.*

### B.2  CHARACTERISTIC EQUATIONS OF FIRST-ORDER DYNAMICS IN BILINEAR GAMES

Latest work of Zhang & Yu (2020) provides the exact and optimal conditions for popular first-order methods to converge in zero-sum bilinear games, if possible. Besides from the exact conditions and the choice of optimal hyperparameters, they also derive the characteristic equation of each first-order dynamics in the zero-sum bilinear games. Since our proofs of theorems in Section 3.2 heavily

---

[1]Actually, $\mathbf{M}$ does not have to be diagonalizable; see Theorem 5.4 and Theorem 5.D4 in Chen (1995).

rely on these characteristic equations, we restate somewhat simplified version of the equations for Equation 9 using our notations.

$$\text{GD}_{\text{Alt}} : \ (\lambda_i - 1)^2 + \eta^2 \sigma_i^2 \lambda_i = 0. \tag{14}$$

$$\text{GD}_{\text{Sim}} : \ (\lambda_i - 1)^2 + \eta^2 \sigma_i^2 = 0. \tag{15}$$

$$\text{PP}_{\text{Sim}} : \ (1/\lambda_i - 1)^2 + \eta^2 \sigma_i^2 = 0. \tag{16}$$

$$\text{EG}_{\text{Alt}} : \ (\lambda_i - 1)^2 + (\eta^2 + 2\eta)\sigma_i^2(\lambda_i - 1) + (\eta^2\sigma_i^2 + \eta^2\sigma_i^4) = 0. \tag{17}$$

$$\text{EG}_{\text{Sim}} : \ (\lambda_i - 1)^2 + 2\eta\sigma_i^2(\lambda_i - 1) + \eta^2\sigma_i^2 + \eta^2\sigma_i^4 = 0. \tag{18}$$

We denote the singular values of matrix $\mathbf{A}$ in Equation 9 by $\sigma_i$. The eigenvalues of each dynamics' Jacobian matrix are denoted by $\lambda_i$. Note that Zhang & Yu (2020) also derive characteristic equations of memory-augmented first-order methods, such as OGD (Popov, 1980) and the momentum method, which we do not cover in this paper.

## C  OMITTED RESULTS

**Proposition 16.** *Alternating GD dynamics $F_{GD_{Alt}}$ with a learning rate $\eta \in (0, 2)$ fails to converge and oscillates around the Nash equilibrium of the game in Equation 8. However, its Lookahead dynamics $G_{LA\text{-}GD_{Alt}}$ with a synchronization period $k \in \mathbb{N}$ and a rate $\alpha \in (0, 1)$ globally converges to the Nash equilibrium if $\left(1 - \frac{\eta^2}{2} + \frac{i\sqrt{4-\eta^2}}{2}\right)^k \neq 1$.*

*Proof.* One can easily check from Equation 2 that the dynamics $F_{\text{GD}_{\text{Alt}}}$ can be written as

$$F_{\text{GD}_{\text{Alt}}}(x_1^{(t)}, x_2^{(t)}) = \begin{bmatrix} 1 & -\eta \\ \eta & 1 - \eta^2 \end{bmatrix} \begin{bmatrix} x_1^{(t)} \\ x_2^{(t)} \end{bmatrix}. \tag{19}$$

Defining $\mathbf{M} \stackrel{\text{def}}{=} \begin{bmatrix} 1 & -\eta \\ \eta & 1 - \eta^2 \end{bmatrix}$, the Lookahead dynamics $G_{\text{LA-GD}_{\text{Alt}}}$ can be written as

$$G_{\text{LA-GD}_{\text{Alt}}}(x_1^{(t)}, x_2^{(t)}) = ((1 - \alpha)\mathbf{I} + \alpha\mathbf{M}^k) \begin{bmatrix} x_1^{(t)} \\ x_2^{(t)} \end{bmatrix}. \tag{20}$$

It follows that the eigenvalues of $\nabla_{\mathbf{x}}G_{\text{LA-GD}_{\text{Alt}}}$ can be written as $1 - \alpha + \alpha\lambda_{\pm}^k$ with $\lambda_{\pm} \stackrel{\text{def}}{=} 1 - \frac{\eta^2}{2} \pm \frac{i\sqrt{4-\eta^2}}{2} \in \lambda(\mathbf{M})$ for any $\eta \in (0, 2)$. However, $1 - \alpha + \alpha\lambda_{\pm}^k$ is an interpolation between two distinct points on $S_1$ since $|\lambda_{\pm}| = 1$ and $\lambda_{\pm}^k \neq 1$, implying $|1 - \alpha + \alpha\lambda_{\pm}^k| < 1$. Therefore, we conclude from Lemma 14 that the iterates of $G_{\text{LA-GD}_{\text{Alt}}}$ converge to the Nash equilibrium $(0, 0)$ of the game with convergence rate $\mathcal{O}(|1 - \alpha + \alpha\lambda_{\pm}^k|^{t/k})$, assuming the amortization of its computation over $k$ forward steps. The proof for oscillation of $F_{\text{GD}_{\text{Alt}}}$ follows from Lemma 15 and can be found in Gidel et al. (2019a). □

**Proposition 17.** *Simultaneous EG Lookahead dynamics $G_{LA\text{-}EG_{Sim}}$ with a learning rate $\eta \in (0, 1)$, a synchronization period $k \in \mathbb{N}$ and a rate $\alpha \in (0, 1)$ globally converges to the Nash equilibrium of Equation 8. Furthermore, the rate of convergence is improved upon its base dynamics $F_{EG_{Sim}}$ if $\Re((1 - \eta^2 + i\eta)^k) < (1 - \eta^2 + \eta^4)^k$ and $\alpha$ is large enough.*

*Proof.* Using simple algebra on Equation 5, the dynamics $F_{\text{EG}_{\text{Sim}}}$ can be written as

$$F_{\text{EG}_{\text{Sim}}}(x_1^{(t)}, x_2^{(t)}) = \begin{bmatrix} 1 - \eta^2 & -\eta \\ \eta & 1 - \eta^2 \end{bmatrix} \begin{bmatrix} x_1^{(t)} \\ x_2^{(t)} \end{bmatrix}. \tag{21}$$

Defining $\mathbf{M} \stackrel{\text{def}}{=} \frac{1}{1+\eta} \begin{bmatrix} 1 - \eta^2 & -\eta \\ \eta & 1 - \eta^2 \end{bmatrix}$, its Lookahead dynamics $G_{\text{LA-EG}_{\text{Sim}}}$ with a synchronization period $k \in \mathbb{N}$ and a rate $\alpha \in (0, 1)$ can be written as

$$G_{\text{LA-EG}_{\text{Sim}}}(x_1^{(t)}, x_2^{(t)}) = ((1 - \alpha)\mathbf{I} + \alpha\mathbf{M}^k) \begin{bmatrix} x_1^{(t)} \\ x_2^{(t)} \end{bmatrix}. \tag{22}$$

It follows that the eigenvalues of $\nabla_{\mathbf{x}} G_{\text{LA-EG}_{\text{Sim}}}$ are $1 - \alpha + \alpha \lambda_\pm^k$ with $\lambda_\pm \overset{\text{def}}{=} 1 - \eta^2 \pm i\eta \in \lambda(\mathbf{M})$. However, $1 - \alpha + \alpha \lambda_\pm^k$ is an interpolation between two distinct points on/inside $S_1$ since $|\lambda_\pm|^k < 1$ for any $\eta \in (0, 1)$. It follows that $|1 - \alpha + \alpha \lambda_\pm^k| < 1$, from which we conclude from Lemma 14 that the iterates of $G_{\text{LA-EG}_{\text{Sim}}}$ converge to the Nash equilibrium $(0, 0)$ of the game with convergence rate $\mathcal{O}(|1 - \alpha + \alpha \lambda_\pm^k|^{t/k})$, assuming the amortization of its computation over $k$ forward steps.

Now we show that the convergence is accelerated upon its base dynamics $F_{\text{EG}_{\text{Sim}}}$ if $\Re((1 - \eta^2 + i\eta)^k) < (1 - \eta^2 + \eta^4)^{2k}$ and $\alpha$ is large enough. Figure 1 (c) intuitively shows that the line segment between $(1, 0)$ and $\lambda_\pm^k$ contains a line segment inside $S_{|\lambda_\pm|^k}$ when $k$ is such that $\Re(\lambda_\pm^k) < |\lambda_\pm^{2k}|$. Therefore, for a large enough $\alpha$, the interpolation $1 - \alpha + \alpha \lambda_\pm^k$ lies inside $S_{|\lambda_\pm|^k}$. This implies that the convergence rate $\mathcal{O}(|1 - \alpha + \alpha \lambda_\pm^k|^{t/k})$ of $G_{\text{LA-EG}_{\text{Sim}}}$ is accelerated upon the rate $\mathcal{O}(|\lambda_\pm|^t)$ of its base dynamics. $\qquad \square$

**Proposition 18** (Equilibrium of Lookahead dynamics). *Let $F$ be a dynamics and $G$ be its associated Lookahead dynamics with a synchronization period $k \in \mathbb{N}$. Then any equilibrium of $F$ is an equilibrium of $G$ and any equilibrium of $G$ is a periodic point of $F$.*

*Proof.* Let $k \in \mathbb{N}$ and $\alpha \in (0, 1)$ be the synchronization period and synchronization rate of $G$, respectively. It is trivial to see that $G(\mathbf{x}^*) = ((1 - \alpha)id + \alpha F^k)(\mathbf{x}^*) = (1 - \alpha)\mathbf{x}^* + \alpha \mathbf{x}^* = \mathbf{x}^*$ if $F(\mathbf{x}^*) = \mathbf{x}^*$. Conversely, one can easily check that $G(\mathbf{x}^*) = (1 - \alpha)\mathbf{x}^* + \alpha F^k(\mathbf{x}^*) = \mathbf{x}^*$ implies $F^k(\mathbf{x}^*) = \mathbf{x}^*$. $\qquad \square$

# D  EXPERIMENTAL DETAILS

We report the actual hyperparameters used for the experiments of Section 5 in TableD.2 and D.3. Furthermore, we also provide the detailed derivations of the theoretically recommended range of synchronization period $k \in \mathbb{N}$.

Table D.2: Hyperparameters used for the experiment on the bilinear game

| Configuration | k (Theorem 5-8) | k | $\alpha$ (Theorem 5-8) | $\alpha$ |
|---|---|---|---|---|
| LA-GD$_{\text{Alt}}^+$ | $\mathbb{N}$ | 25 | $(0, 1)$ | 0.1 |
| LA-GD$_{\text{Alt}}^-$ | $\mathbb{N}$ | 5 | $(0, 1)$ | 0.9 |
| LA-GD$_{\text{Sim}}^+$ | $(18.47, 39.62)$ | 25 | small enough | 0.1 |
| LA-GD$_{\text{Sim}}^-$ | $(18.47, 39.62)$ | 5 | small enough | 0.9 |
| LA-EG$_{\text{Sim}}^+$ | $(16.9, 34.93)$ | 25 | large enough | 0.9 |
| LA-EG$_{\text{Sim}}^-$ | $(16.9, 34.93)$ | 5 | large enough | 0.1 |

Table D.3: Hyperparameters used for the experiment on the nonlinear game

| Configuration | k (Theorem 10, 11) | k | $\alpha$ (Theorem 10,11) | $\alpha$ |
|---|---|---|---|---|
| LA-GD$_{\text{Alt}}^+$ | $(31.16, 93.49)$ | 35 | small enough | 0.1 |
| LA-GD$_{\text{Alt}}^-$ | $(31.16, 93.49)$ | 5 | small enough | 0.9 |
| LA-GD$_{\text{Sim}}^+$ | $(31.6, 94.81)$ | 35 | small enough | 0.1 |
| LA-GD$_{\text{Sim}}^-$ | $(31.6, 94.81)$ | 5 | small enough | 0.9 |
| LA-EG$_{\text{Sim}}^+$ | $(121.76, 365.30)$ | 175 | large enough | 0.9 |
| LA-EG$_{\text{Sim}}^-$ | $(121.76, 365.30)$ | 5 | large enough | 0.1 |

## D.1  DERIVATION OF THEORETICALLY RECOMMENDED RANGE OF $k$ IN EQUATION 12

We plug in $\sigma(\mathbf{A}) = \{1.195, 1.163, 1.094, 1.083, 1.018, 0.999, 0.969, 0.888, 0.879, 0.852\}$ with $\sigma_{\max} = 1.195$ and $\sigma_{\min} = 0.852$ to Theorem 5-8. Then we have

- LA-GD$_{\text{Alt}}$: $\{k \in \mathbb{N} : k \arccos(1 - \frac{0.1^2 \sigma_i^2}{2}) \bmod \pi \neq 0, \forall \sigma_i\}$,
- LA-GD$_{\text{Sim}}$: $\left(\frac{\pi}{2 \arctan 0.08}, \frac{3\pi}{2 \arctan 0.12}\right) = (18.47, 39.62)$,
- LA-EG$_{\text{Sim}}$: $\left(\frac{\pi}{2 \arctan 0.09}, \frac{3\pi}{2 \arctan 0.13}\right) = (16.9, 34.93)$,

which give ranges for $k$ as in Table D.2.

## D.2 Derivation of Theoretically Recommended Range of $k$ in Equation 13

**LA-GD$_{\text{Alt}}$**   From Equation 2, the Jacobian of dynamics $F_{\text{LA-GD}_{\text{Alt}}}$ of Equation 13 can be derived as

$$\nabla_{\mathbf{x}} F_{\text{GD}_{\text{Alt}}}(x_1, x_2) = \begin{bmatrix} 1 & -\eta \\ \eta & 1 - \eta^2 + \eta\epsilon(1 - 3x_2^2) \end{bmatrix} \begin{bmatrix} x_1 \\ x_2 \end{bmatrix}, \tag{23}$$

and it is trivial to see that it has an equilibrium at $(0,0)$. By plugging in $\epsilon = 0.01$ and $\eta = 0.05$, we obtain

$$\nabla_{\mathbf{x}} F_{\text{GD}_{\text{Alt}}}(0,0) = \begin{bmatrix} 1 & -0.05 \\ 0.05 & 0.998 \end{bmatrix} \tag{24}$$

with eigenvalues $\lambda_\pm \stackrel{\text{def}}{=} 0.99 \pm 0.05i$. Note that $|\lambda_\pm| = 1.0003 > 1$ and $\nabla_{\mathbf{x}} F_{\text{GD}_{\text{Alt}}}(0,0)$ has the imaginary conditioning of 1, which implies that the origin is an unstable equilibrium of GD$_{\text{Alt}}$ that can be locally stabilized by a Lookahead dynamics. By plugging in the eigenvalues and $\theta_{\min}(\nabla_{\mathbf{x}} F_{\text{GD}_{\text{Alt}}}(0,0)) = \theta_{\max}(\nabla_{\mathbf{x}} F_{\text{GD}_{\text{Alt}}}(0,0)) = \arctan \frac{0.05}{0.99} = 0.0504$ to Theorem 10, we obtain the theoretically recommended range of $k$ as $(31.16, 93.49)$.

**LA-GD$_{\text{Sim}}$**   From Equation 1, the Jacobian of dynamics $F_{\text{LA-GD}_{\text{Sim}}}$ of Equation 13 can be derived as

$$\nabla_{\mathbf{x}} F_{\text{GD}_{\text{Sim}}}(x_1, x_2) = \begin{bmatrix} 1 & -\eta \\ \eta & 1 + \eta\epsilon(1 - 3x_2^2) \end{bmatrix} \begin{bmatrix} x_1 \\ x_2 \end{bmatrix}, \tag{25}$$

and it is trivial to see that it has an equilibrium at $(0,0)$. By plugging in $\epsilon = 0.01$ and $\eta = 0.05$, we obtain

$$\nabla_{\mathbf{x}} F_{\text{GD}_{\text{Sim}}}(0,0) = \begin{bmatrix} 1 & -0.05 \\ 0.05 & 1.005 \end{bmatrix} \tag{26}$$

with eigenvalues $\lambda_\pm \stackrel{\text{def}}{=} 1.0025 \pm 0.0499i$. Note that $|\lambda_\pm| = 1.0037 > 1$ and $\nabla_{\mathbf{x}} F_{\text{GD}_{\text{Sim}}}(0,0)$ has the imaginary conditioning of 1, which implies that the origin is an unstable equilibrium of GD$_{\text{Alt}}$ that can be locally stabilized by a Lookahead dynamics. By plugging in the eigenvalues and $\theta_{\min}(\nabla_{\mathbf{x}} F_{\text{GD}_{\text{Sim}}}(0,0)) = \theta_{\max}(\nabla_{\mathbf{x}} F_{\text{GD}_{\text{Sim}}}(0,0)) = \arctan \frac{0.0499}{1.0025} = 0.0497$ to Theorem 10, we obtain the theoretically recommended range of $k$ as $(31.6, 94.81)$.

**LA-EG$_{\text{Sim}}$**   From Equation 5, the dynamics $F_{\text{LA-EG}_{\text{Sim}}}$ of Equation 13 can be derived as

$$\begin{bmatrix} x_1' \\ x_2' \end{bmatrix} = F_{\text{EG}_{\text{Sim}}}(x_1, x_2) = \begin{bmatrix} x_1 - \eta\tilde{x}_2 \\ x_2 + \eta(\tilde{x}_1 + \epsilon(\tilde{x}_2 - \tilde{x}_2^3)) \end{bmatrix}, \quad \text{where} \tag{27}$$

$$\begin{bmatrix} \tilde{x}_1 \\ \tilde{x}_2 \end{bmatrix} = \begin{bmatrix} x_1 - \eta x_2 \\ x_2 + \eta(x_1 + \epsilon(x_2 - x_2^3)) \end{bmatrix}. \tag{28}$$

By computing the derivatives with $x_1 = 0, x_2 = 0$ and $\epsilon = 0.01, \eta = 0.05$, we obtain

$$\nabla_{\mathbf{x}} F_{\text{EG}_{\text{Sim}}}(0,0) = \begin{bmatrix} 0.9975 & -0.05 \\ 0.05 & 0.9005 \end{bmatrix} \tag{29}$$

with eigenvalues $\lambda_\pm = 0.949 \pm 0.0122i$. Note that $|\lambda_\pm| = 0.949 < 1$ and $\nabla_{\mathbf{x}} F_{\text{EG}_{\text{Sim}}}(0,0)$ has the imaginary conditioning of 1, which implies that the origin is an stable equilibrium of EG$_{\text{Sim}}$ whose local convergence can be accelerated by a Lookahead dynamics. By plugging in the eigenvalues and $\theta_{\min}(\nabla_{\mathbf{x}} F_{\text{EG}_{\text{Sim}}}(0,0)) = \theta_{\max}(\nabla_{\mathbf{x}} F_{\text{EG}_{\text{Sim}}}(0,0)) = \arctan \frac{0.0122}{0.949} = 0.0129$ to Theorem 11, we obtain the theoretically recommended range of $k$ as $(121.76, 365.30)$.

# E PROOFS

## E.1 PROOF OF PROPOSITION 1

*Proof.* One can easily check from Equation 1 that the dynamics $F_{\text{GD}_{\text{Sim}}}$ can be written as

$$F_{\text{GD}_{\text{Sim}}}(x_1^{(t)}, x_2^{(t)}) = \begin{bmatrix} 1 & -\eta \\ \eta & 1 \end{bmatrix} \begin{bmatrix} x_1^{(t)} \\ x_2^{(t)} \end{bmatrix}. \tag{30}$$

Defining $\mathbf{M} \overset{\text{def}}{=} \begin{bmatrix} 1 & -\eta \\ \eta & 1 \end{bmatrix}$, its Lookahead dynamics $G_{\text{LA-GD}_{\text{Sim}}}$ can be written as

$$G_{\text{LA-GD}_{\text{Sim}}}(x_1^{(t)}, x_2^{(t)}) = ((1-\alpha)\mathbf{I} + \alpha\mathbf{M}^k) \begin{bmatrix} x_1^{(t)} \\ x_2^{(t)} \end{bmatrix}. \tag{31}$$

It follows that the eigenvalues of $\nabla_{\mathbf{x}} G_{\text{LA-GD}_{\text{Sim}}}$ can be written as $1 - \alpha + \alpha\lambda_{\pm}^k$ with $\lambda_{\pm} \overset{\text{def}}{=} 1 \pm i\eta \in \lambda(\mathbf{M})$. Assuming $\Re((1+i\eta)^k) < 1$, the line segment between $(1, 0)$ and $\lambda_{\pm}^k$ contains a line segment inside $S_1$ as in Figure 1 (b). Therefore, for a small enough $\alpha$, the interpolation $1 - \alpha + \alpha\lambda_{\pm}^k$ lies inside $S_1$, implying $|1 - \alpha + \alpha\lambda_{\pm}^k| < 1$. We thus conclude from Lemma 15 that the iterates of $G_{\text{LA-GD}_{\text{Sim}}}$ converge to the Nash equilibrium $(0, 0)$ of the game. The proof for divergence of $F_{\text{GD}_{\text{Sim}}}$ follows from Lemma 15 and can be found in Gidel et al. (2019a). □

## E.2 PROOF OF PROPOSITION 2

*Proof.* Using simple algebra on Equation 4, the dynamics $F_{\text{PP}_{\text{Sim}}}$ can be written as

$$F_{\text{PP}_{\text{Sim}}}(x_1^{(t)}, x_2^{(t)}) = \frac{1}{1+\eta} \begin{bmatrix} 1 & -\eta \\ \eta & 1 \end{bmatrix} \begin{bmatrix} x_1^{(t)} \\ x_2^{(t)} \end{bmatrix}. \tag{32}$$

Defining $\mathbf{M} \overset{\text{def}}{=} \frac{1}{1+\eta} \begin{bmatrix} 1 & -\eta \\ \eta & 1 \end{bmatrix}$, its Lookahead dynamics $G_{\text{LA-PP}_{\text{Sim}}}$ with a synchronization period $k \in \mathbb{N}$ and a rate $\alpha \in (0, 1)$ can be written as

$$G_{\text{LA-PP}_{\text{Sim}}}(x_1^{(t)}, x_2^{(t)}) = ((1-\alpha)\mathbf{I} + \alpha\mathbf{M}^k) \begin{bmatrix} x_1^{(t)} \\ x_2^{(t)} \end{bmatrix}. \tag{33}$$

It follows that the eigenvalues of $\nabla_{\mathbf{x}} G_{\text{LA-PP}_{\text{Sim}}}$ are $1 - \alpha + \alpha\lambda_{\pm}^k$ with $\lambda_{\pm} \overset{\text{def}}{=} \frac{1 \pm i\eta}{1+\eta^2} \in \lambda(\mathbf{M})$. We know that $1 - \alpha + \alpha\lambda_{\pm}^k$ is an interpolation between two distinct points on/inside $S_1$ since $|\lambda_{\pm}|^k < 1$ for any $\eta \in (0, 1)$. It follows that $|1 - \alpha + \alpha\lambda_{\pm}^k| < 1$, from which we conclude from Lemma 14 that the iterates of $G_{\text{LA-PP}_{\text{Sim}}}$ converge to the Nash equilibrium $(0, 0)$ of the game with convergence rate $\mathcal{O}(|1 - \alpha + \alpha\lambda_{\pm}^k|^{t/k})$, assuming the amortization of its computation over $k$ forward steps.

Now we show that the convergence is accelerated upon the base dynamics $F_{\text{PP}_{\text{Sim}}}$ if $\Re((1 + i\eta)^k) < (1 + \eta^2)^k$ and $\alpha$ is large enough. Figure 1 (c) intuitively shows that the line segment between $(1, 0)$ and $\lambda_{\pm}^k$ contains a line segment inside $S_{|\lambda_{\pm}|^k}$ when $k$ is such that $\Re(\lambda_{\pm}^k) < |\lambda_{\pm}^{2k}|$. Therefore, the interpolation $1 - \alpha + \alpha\lambda_{\pm}^k$ lies inside $S_{|\lambda_{\pm}|^k}$ for a large enough $\alpha$. This implies that the convergence rate $\mathcal{O}(|1 - \alpha + \alpha\lambda_{\pm}^k|^{t/k})$ of $G_{\text{LA-PP}_{\text{Sim}}}$ is accelerated upon the rate $\mathcal{O}(|\lambda_{\pm}|^t)$ of its base dynamics. □

## E.3 PROOF OF LEMMA 3

*Proof.* We prove each of the cases in their order.

**Case** $\rho(\mathbf{X}) = 1$. Assume that $\lambda_{\max}^k \neq 1$ for any $\lambda_{\max} \in \lambda_{\max}(\mathbf{X})$. Then we can immediately conclude $\rho(f(\mathbf{X})) < 1$ since $1 - \alpha + \alpha\lambda_i^k \in \lambda(f(\mathbf{X}))$ is an interpolation between two distinct points $(1, 0)$ and $\lambda_i^k$ on/inside $S_1$ for any $\lambda_i \in \lambda(\mathbf{X})$.

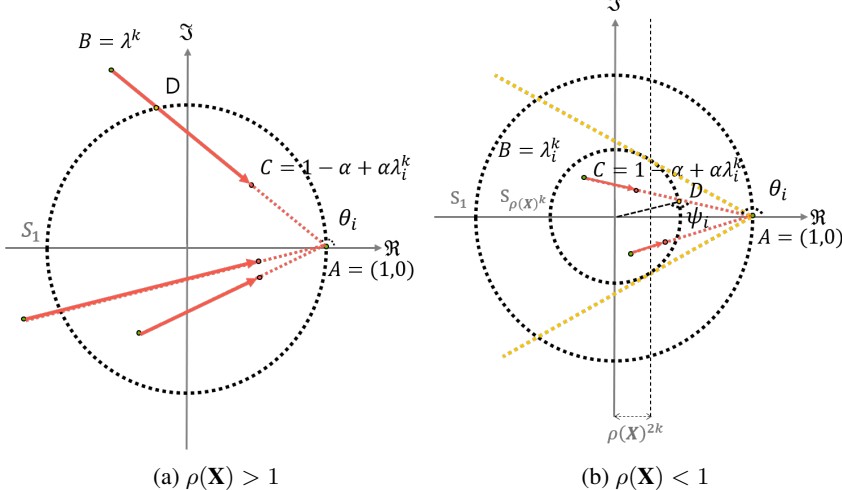

(a) $\rho(\mathbf{X}) > 1$        (b) $\rho(\mathbf{X}) < 1$

Figure 4: Visualized eigenvalues of $(1 - \alpha)\mathbf{I} + \alpha\mathbf{X}^k$.

**Case $\rho(\mathbf{X}) > 1$.** Assume that $\Re(\lambda^k) < 1$ for any $\lambda \in \lambda_{\geq 1}(\mathbf{X})$. Then for each $\lambda \in \lambda_{\geq 1}$, $\lambda^k$ can be visualized as point B in Figure 4 (a), where the existence of point D is guaranteed by $\overline{\Re}(\lambda) < 1$. It is easy to see from the figure that

$$\left\|\overline{AC}\right\| = \alpha|\lambda^k - 1| < 2\cos(\pi - \theta(\lambda)) = \left\|\overline{AD}\right\| \tag{34}$$

is sufficient to place $1 - \alpha + \alpha\lambda^k$ inside $S_1$. Furthermore, for any $\lambda \in \lambda(\mathbf{X})$ such that $|\lambda| < 1$, $1 - \alpha + \alpha\lambda^k$ lies inside $S_1$ since $1 - \alpha + \alpha\lambda^k$ is an interpolation between two distinct points on/inside $S_1$. Therefore we conclude $\rho(f(\mathbf{X})) < 1$.

**Case $\rho(\mathbf{X}) < 1$.** Assume that $\Re(\lambda^k) < \rho(\mathbf{X})^{2k}$ for any $\lambda \in \lambda_{\max}(\mathbf{X})$. Then for any $\lambda_i \in \lambda(\mathbf{X})$, $\lambda_i^k$ can be visualized as point B in Figure 4 (b) since the existence of point D is guaranteed by $\Re(\lambda^k) < \rho(\mathbf{X})^{2k}$ and $\sin(\phi(\lambda_i)) = \sin(\theta(\lambda_i))/\rho(\mathbf{X})^k$ follows from the law of sines. Therefore we can intuitively see from the figure that

$$\left\|\overline{BC}\right\| = (1 - \alpha)|\lambda_i^k - 1| < 2\rho(\mathbf{X})^k\cos(\pi - \phi(\lambda_i)) = \left\|\overline{BD}\right\| \tag{35}$$

is sufficient to place $1 - \alpha + \alpha\lambda_i^k$ inside $S_{\rho(X)^k}$, concluding the proof. $\qquad\square$

### E.4    PROOF OF LEMMA 4

*Proof.* Let us denote $\theta_{\min} \stackrel{\text{def}}{=} \theta_{\min}(S)$, $\theta_{\max} \stackrel{\text{def}}{=} \theta_{\max}(S)$ for brevity and let $k \in \mathbb{N}$ be such that $k \in \left(\frac{\pi}{2\theta_{\min}}, \frac{3\pi}{2\theta_{\max}}\right)$. Then we have $k\theta_{\min} \in \left(\frac{\pi}{2}, \frac{3\pi\theta_{\min}}{2\theta_{\max}}\right) \subseteq \left(\frac{\pi}{2}, \frac{3\pi}{2}\right)$ and $k\theta_{\max} \in \left(\frac{\pi\theta_{\max}}{2\theta_{\min}}, \frac{3\pi}{2}\right) \subseteq \left(\frac{\pi}{2}, \frac{3\pi}{2}\right)$, which implies $\Re(\lambda_i^k) < 0$ for any $\lambda_i \in S$ such that $\Im(\lambda_i) > 0$. Since every element of $S$ has its conjugate pair in $S$ by the assumption, we conclude $\Re(\lambda_i^k) < 0$ for any $\lambda_i \in S$.

Now we show that the existence of $k \in \mathbb{N}$ such that $k \in \left(\frac{\pi}{2\theta_{\min}}, \frac{3\pi}{2\theta_{\max}}\right)$ is guaranteed for a small enough $\eta > 0$ when $\frac{\Im_{\max}(S)}{\Im_{\min}(S)} < 3$. Using simple algebra, we can see that $\theta_{\max} < f(\theta_{\min})$ for $f : \mathbb{R} \to \mathbb{R}$ defined by $f(x) = \frac{3\pi x}{\pi + 2x}$ is equivalent to $\frac{\pi}{2\theta_{\min}} - \frac{3\pi}{2\theta_{\max}} > 1$, implying nonempty $\mathbb{N} \cap \left(\frac{\pi}{2\theta_{\min}}, \frac{3\pi}{2\theta_{\max}}\right)$. Therefore it suffices to show that $\theta_{\max} < f(\theta_{\min})$ holds for a small enough $\eta > 0$ when $\frac{\Im_{\max}(S)}{\Im_{\min}(S)} < 3$.

Let us define a function $H : \mathbb{R} \to \mathbb{R}$ given by

$$H(\eta) \stackrel{\text{def}}{=} \left(1 + \frac{2\theta_{\max}^{+}}{\pi}\right)\left(\frac{1 + \eta\Re_{\max}(S)}{1 + \eta\Re_{\min}(S)}\right)\left(\frac{1 + 2\sec\theta_{\min}^{-}}{1 + 2\sec\theta_{\max}^{+}} + b\right), \tag{36}$$

where $\theta_{\min}^{-} \stackrel{\text{def}}{=} \arctan \frac{\eta \Im_{\min}(S)}{1+\eta \Re_{\max}(S)}$, $\theta_{\max}^{+} \stackrel{\text{def}}{=} \arctan \frac{\eta \Im_{\max}(S)}{1+\eta \Re_{\min}(S)}$ and $b \stackrel{\text{def}}{=} \frac{(1+2 \sec \theta_{\min}^{-}) \tan^4 \theta_{\max}^{+}}{540}$.

We show that the inequality

$$\frac{\Im_{\max}(S)}{\Im_{\min}(S)} < \frac{3}{H(\eta)} \tag{37}$$

implies $\theta_{\max} < f(\theta_{\min})$ and conclude the proof by showing that there exists a small enough $\eta > 0$ such that satisfies Equation 37 when $\frac{\Im_{\max}(S)}{\Im_{\min}(S)} < 3$.

Note that the inequalities $\theta_{\min}^{-} \leq \theta_{\min}$ and $\theta_{\max} \leq \theta_{\max}^{+}$ directly follow from the definitions of $\theta_{\min}^{-}$ and $\theta_{\max}^{+}$. Furthermore, using the Shafer-type double inequalities (Mortici & Srivastava, 2014) for $\arctan(\cdot)$, we obtain

$$\theta_{\min}^{-} \geq \frac{3 \tan \theta_{\min}^{-}}{1 + 2\sqrt{1 + \tan^2 \theta_{\min}^{-}}} = \frac{3\eta \Im_{\min}}{(1 + \eta \Re_{\max})(1 + 2 \sec \theta_{\min}^{-})}, \tag{38}$$

$$\theta_{\max}^{+} \leq \frac{3 \tan \theta_{\max}^{+}}{1 + 2\sqrt{1 + \tan^2 \theta_{\max}^{+}}} + \frac{1}{180} \tan^5 \theta_{\max}^{+} \tag{39}$$

$$= \frac{3\eta \Im_{\max}}{(1 + \eta \Re_{\min})(1 + 2 \sec \theta_{\max}^{+})} + \frac{\eta \Im_{\max} \tan^4 \theta_{\max}^{+}}{180(1 + \eta \Re_{\min})}, \tag{40}$$

from which follows that

$$\frac{\theta_{\max}}{\theta_{\min}} \leq \frac{\theta_{\max}^{+}}{\theta_{\min}^{-}} = \frac{\Im_{\max}(S)}{\Im_{\min}(S)} \left( \frac{1 + \eta \Re_{\max}(S)}{1 + \eta \Re_{\min}(S)} \right) \left( \frac{1 + 2 \sec \theta_{\min}^{-}}{1 + 2 \sec \theta_{\max}^{+}} + b \right). \tag{41}$$

However, assuming inequality 37, we can derive

$$\frac{\Im_{\max}(S)}{\Im_{\min}(S)} \left( \frac{1 + \eta \Re_{\max}(S)}{1 + \eta \Re_{\min}(S)} \right) \left( \frac{1 + 2 \sec \theta_{\min}^{-}}{1 + 2 \sec \theta_{\max}^{+}} + b \right) < \frac{3\pi}{\pi + 2\theta_{\max}^{+}} = \frac{f(\theta_{\max}^{+})}{\theta_{\max}^{+}}. \tag{42}$$

Furthermore, since $f'(x) = \frac{3\pi^2}{(\pi + 2x)^2}$, we know that $f$ is both concave and monotonically increasing. Hence it follows that

$$\frac{f(\theta_{\max}^{+})}{\theta_{\max}^{+}} < \frac{f(\theta_{\min})}{\theta_{\min}}, \tag{43}$$

from which we obtain $\theta_{\max} < f(\theta_{\min})$ by combining Equation 41-43.

Finally, we prove that Equation 37 holds for a small enough $\eta > 0$ when $\frac{\Im_{\max}(S)}{\Im_{\min}(S)} < 3$. Assume $\frac{\Im_{\max}(S)}{\Im_{\min}(S)} < 3$ and let $\epsilon \stackrel{\text{def}}{=} 3 - \frac{\Im_{\max}(S)}{\Im_{\min}(S)} > 0$. By the continuity of $\frac{3}{H(\cdot)}$ at $\eta = 0$ and the fact that $H(0) = 1$, there exists $\delta > 0$ such that $|3 - \frac{3}{H(\eta)}| < \epsilon$ holds for any $\eta \in (0, \delta)$. Therefore we have $\frac{\Im_{\max}(S)}{\Im_{\min}(S)} = 3 - \epsilon < \frac{3}{H(\eta)}$ for any $\eta \in (0, \delta)$, concluding the proof. □

### E.5 PROOF OF THEOREM 5

*Proof.* From Equation 2, the dynamics $F_{\text{GD}_{\text{Alt}}}$ of Equation 11 can be derived as

$$F_{\text{GD}_{\text{Alt}}}(\mathbf{x}_1^{(t)}, \mathbf{x}_2^{(t)}) = \begin{bmatrix} \mathbf{I}_r & -\eta \Sigma_r \\ \eta \Sigma_r & \mathbf{I}_r - \eta^2 \Sigma_r^2 \end{bmatrix} \begin{bmatrix} \mathbf{x}_1^{(t)} \\ \mathbf{x}_2^{(t)} \end{bmatrix}. \tag{44}$$

Defining $\mathbf{M} \stackrel{\text{def}}{=} \begin{bmatrix} \mathbf{I}_r & -\eta \Sigma_r \\ \eta \Sigma_r & \mathbf{I}_r - \eta^2 \Sigma_r^2 \end{bmatrix}$, its Lookahead dynamics $G_{\text{LA-GD}_{\text{Alt}}}$ with a synchronization period $k \in \mathbb{N}$ and a rate $\alpha \in (0, 1)$ can be written as

$$G_{\text{LA-GD}_{\text{Alt}}}(\mathbf{x}_1^{(t)}, \mathbf{x}_2^{(t)}) = ((1 - \alpha)\mathbf{I} + \alpha \mathbf{M}^k) \begin{bmatrix} \mathbf{x}_1^{(t)} \\ \mathbf{x}_2^{(t)} \end{bmatrix}. \tag{45}$$

Together with Equation 14, we can see that eigenvalues of $\nabla_{\mathbf{x}} G_{\text{LA-GD}_{\text{Alt}}}$ can be written as $1 - \alpha + \alpha \lambda_{\pm i}^k$ with $\lambda_{\pm i} \stackrel{\text{def}}{=} 1 - \frac{\eta^2 \sigma_i^2}{2} \pm i\eta\sigma_i \sqrt{1 - \frac{\eta^2 \sigma_i^2}{4}} \in \lambda(\mathbf{M})$ for any $\eta \in \left(0, \frac{2}{\sigma_{\max}}\right)$. In the meanwhile, simple calculation gives us $|\lambda_{\pm i}| = 1$, which implies $\rho(\mathbf{M}) = 1$. Now assume $k \in \mathbb{N}$ is such that $k \arccos(1 - \frac{\eta^2 \sigma_i^2}{2}) \mod 2\pi \neq 0$ for any $\sigma_i$. Then it follows that $\lambda_{\pm i}^k \neq 1$ for any $\lambda_{\pm i} \in \lambda(\mathbf{M})$, from which we obtain $\rho(\nabla_{\mathbf{x}} G_{\text{LA-GD}_{\text{Alt}}}) < 1$ from Lemma 3. It follows from Lemma 15 that the iterates converge to the origin, and we conclude the proof by observing that the transformations $\mathbf{x}_1 \mapsto \mathbf{U}[\mathbf{x}_1; \mathbf{0}_{m-r}] + \mathbf{x}_1^*$ and $\mathbf{x}_2 \mapsto \mathbf{V}[\mathbf{x}_2; \mathbf{0}_{n-r}] + \mathbf{x}_2^*$ of $(\mathbf{0}, \mathbf{0}) \in \mathbb{R}^r \times \mathbb{R}^r$ gives $(\mathbf{x}_1^*, \mathbf{x}_2^*) \in \mathbb{R}^m \times \mathbb{R}^n$, which is a Nash equilibrium of Equation 9. $\qquad\square$

### E.6 PROOF OF THEOREM 6

*Proof.* From Equation 1, the dynamics $F_{\text{GD}_{\text{Sim}}}$ of Equation 11 can be derived as

$$F_{\text{GD}_{\text{Sim}}}(\mathbf{x}_1^{(t)}, \mathbf{x}_2^{(t)}) = \begin{bmatrix} \mathbf{I}_r & -\eta\Sigma_r \\ \eta\Sigma_r & \mathbf{I}_r \end{bmatrix} \begin{bmatrix} \mathbf{x}_1^{(t)} \\ \mathbf{x}_2^{(t)} \end{bmatrix}. \tag{46}$$

Let us define $\mathbf{J} \stackrel{\text{def}}{=} \begin{bmatrix} \mathbf{0}_r & \Sigma_r \\ -\Sigma_r & \mathbf{0}_r \end{bmatrix}$ and $\mathbf{M} \stackrel{\text{def}}{=} \mathbf{I} - \eta\mathbf{J}$. Then its Lookahead dynamics $G_{\text{LA-GD}_{\text{Sim}}}$ with a synchronization period $k \in \mathbb{N}$ and a rate $\alpha \in (0, 1)$ can be written as

$$G_{\text{LA-GD}_{\text{Sim}}}(\mathbf{x}_1^{(t)}, \mathbf{x}_2^{(t)}) = ((1 - \alpha)\mathbf{I} + \alpha\mathbf{M}^k) \begin{bmatrix} \mathbf{x}_1^{(t)} \\ \mathbf{x}_2^{(t)} \end{bmatrix}. \tag{47}$$

Together with Equation 15, we can see that the eigenvalues of $\nabla_{\mathbf{x}} G_{\text{LA-GD}_{\text{Sim}}}$ can be written as $1 - \alpha + \alpha \lambda_{\pm i}^k$ with $\lambda_{\pm i} \stackrel{\text{def}}{=} 1 \pm i\eta\sigma_i \in \lambda(\mathbf{M})$. In the meanwhile, one can easily see that $|\lambda_{\pm i}| > 1$, implying $\rho(\mathbf{M}) > 1$. Now assume that $k \in \left(\frac{\pi}{2\arctan \eta\sigma_{\min}}, \frac{3\pi}{2\arctan \eta\sigma_{\max}}\right)$. Then since $\tan\theta_{\min}(\lambda(\mathbf{M})) = \eta\sigma_{\min}$ and $\tan\theta_{\max}(\lambda(\mathbf{M})) = \eta\sigma_{\max}$, we have $k \in \left(\frac{\pi}{2\theta_{\min}(\lambda(\mathbf{M}))}, \frac{3\pi}{2\theta_{\max}(\lambda(\mathbf{M}))}\right)$. It follows from Lemma 4 that $\Re(\lambda_{\pm i}^k) < 0$ for any $\lambda_{\pm i} \in \lambda(\mathbf{M})$, and the existence of $k$ is guaranteed for a small enough $\eta$ when $\frac{\Im_{\max}(\lambda(\mathbf{M}))}{\Im_{\min}(\lambda(\mathbf{M}))} = \frac{\sigma_{\max}}{\sigma_{\min}} < 3$. Then it follows from Lemma 3 that $\rho(\nabla_{\mathbf{x}} G_{\text{LA-GD}_{\text{Sim}}}) < 1$ holds for a small enough $\alpha$. Therefore, by Lemma 15, the iterates converge to the origin, and we conclude the proof by observing that the transformations $\mathbf{x}_1 \mapsto \mathbf{U}[\mathbf{x}_1; \mathbf{0}_{m-r}] + \mathbf{x}_1^*$ and $\mathbf{x}_2 \mapsto \mathbf{V}[\mathbf{x}_2; \mathbf{0}_{n-r}] + \mathbf{x}_2^*$ of $(\mathbf{0}, \mathbf{0}) \in \mathbb{R}^r \times \mathbb{R}^r$ gives $(\mathbf{x}_1^*, \mathbf{x}_2^*) \in \mathbb{R}^m \times \mathbb{R}^n$, which is a Nash equilibrium of Equation 9. $\qquad\square$

### E.7 PROOF OF THEOREM 7

*Proof.* From Equation 4, the dynamics $F_{\text{PP}_{\text{Sim}}}$ of Equation 11 can be derived as

$$F_{\text{PP}_{\text{Sim}}}(\mathbf{x}_1^{(t)}, \mathbf{x}_2^{(t)}) = \begin{bmatrix} \mathbf{I}_r & \eta\Sigma_r \\ -\eta\Sigma_r & \mathbf{I}_r \end{bmatrix}^{-1} \begin{bmatrix} \mathbf{x}_1^{(t)} \\ \mathbf{x}_2^{(t)} \end{bmatrix}. \tag{48}$$

Let us define $\mathbf{J} \stackrel{\text{def}}{=} \begin{bmatrix} \mathbf{0}_r & \Sigma_r \\ -\Sigma_r & \mathbf{0}_r \end{bmatrix}$ and $\mathbf{M} \stackrel{\text{def}}{=} (\mathbf{I} + \eta\mathbf{J})^{-1}$. Then its Lookahead dynamics $G_{\text{LA-GD}_{\text{Sim}}}$ with a synchronization period $k \in \mathbb{N}$ and a rate $\alpha \in (0, 1)$ can be written as

$$G_{\text{LA-PP}_{\text{Sim}}}(\mathbf{x}_1^{(t)}, \mathbf{x}_2^{(t)}) = ((1 - \alpha)\mathbf{I} + \alpha\mathbf{M}^k) \begin{bmatrix} \mathbf{x}_1^{(t)} \\ \mathbf{x}_2^{(t)} \end{bmatrix}. \tag{49}$$

Together with Equation 16, we can see that the eigenvalues of $\nabla_{\mathbf{x}} G_{\text{LA-EG}_{\text{Sim}}}$ can be written as $1 - \alpha + \alpha \lambda_{\pm i}^k$ with $\lambda_{\pm i} \stackrel{\text{def}}{=} \frac{1 + i\eta\sigma_i}{1 + \eta^2 \sigma_i^2} \in \lambda(\mathbf{M})$. In the meanwhile, we can easily see that $|\lambda_{\pm i}| < 1$ holds for any $\eta > 0$. Therefore, $1 - \alpha + \alpha \lambda_{\pm i}^k$ is an interpolation between two distinct points $(1, 0)$ and $\lambda_{\pm i}^k$ on/inside $S_1$, implying $\rho(\nabla_{\mathbf{x}} G_{\text{LA-PP}_{\text{Sim}}}) < 1$. Hence it follows from Lemma 15 that the iterates converges to the origin. However, the transformations $\mathbf{x}_1 \mapsto \mathbf{U}[\mathbf{x}_1; \mathbf{0}_{m-r}] + \mathbf{x}_1^*$ and

$\mathbf{x}_2 \mapsto \mathbf{V}\left[\mathbf{x}_2; \mathbf{0}_{n-r}\right] + \mathbf{x}_2^*$ of $(\mathbf{0}, \mathbf{0}) \in \mathbb{R}^r \times \mathbb{R}^r$ gives $(\mathbf{x}_1^*, \mathbf{x}_2^*) \in \mathbb{R}^m \times \mathbb{R}^n$, which is a Nash equilibrium of Equation 9.

Now we show that $G_{\text{LA-PP}_{\text{Sim}}}$ can accelerate the convergence upon its base dynamics $F_{\text{PP}_{\text{Sim}}}$. Assume $k \in \left( \frac{\pi}{2 \arctan \eta \sigma_{\min}}, \frac{3\pi}{2 \arctan \eta \sigma_{\min}} \right)$. Note that $\mathbf{M}^{-1}$ shares the same eigenvalues with the Jacobian of $F_{\text{GD}_{\text{Sim}}}$. Therefore we have $\tan \theta_{\min}(\lambda_{\max}(\mathbf{M}^{-1})) = \tan \theta_{\max}(\lambda_{\max}(\mathbf{M}^{-1})) = \eta \sigma_{\min}$ , which implies $k \in \left( \frac{\pi}{2\theta_{\min}(\lambda_{\max}(\mathbf{M}^{-1}))}, \frac{3\pi}{2\theta_{\max}(\lambda_{\max}(\mathbf{M}^{-1}))} \right)$. Then it follows from Lemma 4 that $\Re(\lambda_{\pm i}^{-k}) < 0$ for any $\lambda_{\pm i}^{-1} \in \lambda_{\max}(\mathbf{M}^{-1})$, and the existence of $k \in \mathbb{N}$ is guaranteed for a small enough $\eta$. Then we have $\Re(\lambda_{\pm i}^k) < 0$ for any $\lambda_{\pm i} \in \lambda(\mathbf{M})$ since the reciprocal of a complex number preserves the sign of the real part. Hence it follows from Lemma 3 that $\rho(\nabla_{\mathbf{x}} G_{\text{LA-PP}_{\text{Sim}}}) < \rho(\mathbf{M})^k$ holds for a large enough $\alpha$. We conclude the proof by noting that the convergence rate $\mathcal{O}(\rho(\nabla_{\mathbf{x}} G_{\text{LA-PP}_{\text{Sim}}})^{\frac{t}{k}})$ of $G_{\text{LA-PP}_{\text{Sim}}}$ provided by Lemma 15 is faster than the rate $\mathcal{O}(\rho(\mathbf{M})^t)$ of $F_{\text{PP}_{\text{Sim}}}$, assuming amortization of computations over $k$ forward steps. $\square$

### E.8 PROOF OF THEOREM 8

*Proof.* From Equation 5, the dynamics $F_{\text{EG}_{\text{Sim}}}$ of Equation 11 can be derived as

$$F_{\text{EG}_{\text{Sim}}}(\mathbf{x}_1^{(t)}, \mathbf{x}_2^{(t)}) = \begin{bmatrix} \mathbf{I}_r - \eta \Sigma_r^2 & -\eta \Sigma_r \\ \eta \Sigma_r & \mathbf{I}_r - \eta \Sigma_r^2 \end{bmatrix} \begin{bmatrix} \mathbf{x}_1^{(t)} \\ \mathbf{x}_2^{(t)} \end{bmatrix}. \tag{50}$$

Let us define $\mathbf{J} \stackrel{\text{def}}{=} \begin{bmatrix} \Sigma_r^2 & \Sigma_r \\ -\Sigma_r & \Sigma_r^2 \end{bmatrix}$ and $\mathbf{M} \stackrel{\text{def}}{=} \mathbf{I} - \eta \mathbf{J}$. Then its Lookahead dynamics $G_{\text{LA-EG}_{\text{Sim}}}$ with a synchronization period $k \in \mathbb{N}$ and a rate $\alpha \in (0, 1)$ can be written as

$$G_{\text{LA-EG}_{\text{Sim}}}(\mathbf{x}_1^{(t)}, \mathbf{x}_2^{(t)}) = ((1-\alpha)\mathbf{I} + \alpha \mathbf{M}^k) \begin{bmatrix} \mathbf{x}_1^{(t)} \\ \mathbf{x}_2^{(t)} \end{bmatrix}. \tag{51}$$

Together with Equation 18, we can see that the eigenvalues of $\nabla_{\mathbf{x}} G_{\text{LA-EG}_{\text{Sim}}}$ can be written as $1 - \alpha + \alpha \lambda_{\pm i}^k$ with $\lambda_{\pm i} \stackrel{\text{def}}{=} 1 - \eta \sigma_i \pm i\eta \sigma_i \in \lambda(\mathbf{M})$. In the meanwhile, we can easily see that $|\lambda_{\pm i}| < 1$ for any $\eta \in \left( 0, \frac{1}{\sigma_{\max}} \right)$, implying $\rho(\mathbf{M}) < 1$. Therefore, $1 - \alpha + \alpha \lambda_{\pm i}^k$ is an interpolation between two distinct points $(1, 0)$ and $\lambda_{\pm i}^k$ on/inside $S_1$, implying $\rho(\nabla_{\mathbf{x}} G_{\text{LA-EG}_{\text{Sim}}}) < 1$. Hence it follows from Lemma 15 that the iterates converges to the origin. However, the transformations $\mathbf{x}_1 \mapsto \mathbf{U}\left[\mathbf{x}_1; \mathbf{0}_{m-r}\right] + \mathbf{x}_1^*$ and $\mathbf{x}_2 \mapsto \mathbf{V}\left[\mathbf{x}_2; \mathbf{0}_{n-r}\right] + \mathbf{x}_2^*$ on $(\mathbf{0}, \mathbf{0}) \in \mathbb{R}^r \times \mathbb{R}^r$ gives $(\mathbf{x}_1^*, \mathbf{x}_2^*) \in \mathbb{R}^m \times \mathbb{R}^n$, which is a Nash equilibrium of Equation 9.

Now we show that $G_{\text{LA-EG}_{\text{Sim}}}$ can accelerate the convergence upon its base dynamics $F_{\text{EG}_{\text{Sim}}}$. Assume $k \in \left( \frac{\pi}{2 \arctan \frac{\eta \sigma_{\min}}{1 - \eta \sigma_{\min}}}, \frac{3\pi}{2 \arctan \frac{\eta \sigma_{\min}}{1 - \eta \sigma_{\min}}} \right)$ and $\eta \in \left( 0, \frac{1}{2\sigma_{\max}} \right)$. Note that $|\lambda_i|^2 = 2\eta^2 (\sigma_i - \frac{1}{2\eta})^2 + \frac{1}{2}$ holds for each $\lambda_{\pm i} \in \lambda(\mathbf{M})$. This implies $\lambda_{\max}(\mathbf{M}) = \{1 - \eta \sigma_{\min} \pm i\eta \sigma_{\min}\}$ for any $\eta \in \left( 0, \frac{1}{2\sigma_{\max}} \right)$, hence $k \in \left( \frac{\pi}{2\theta_{\min}(\lambda_{\max}(\mathbf{M}))}, \frac{3\pi}{2\theta_{\max}(\lambda_{\max}(\mathbf{M}))} \right)$. It follows from Lemma 4 that $\Re(\lambda_{\pm i}^k) < 0$ holds for any $\lambda_{\pm i} \in \lambda(\mathbf{M})$, and the existence of $k$ is guaranteed for a small enough $\eta$. Then by Lemma 3 we have $\rho(\nabla_{\mathbf{x}} G_{\text{LA-EG}_{\text{Sim}}}) < \rho(\mathbf{M})^k$ for a large enough $\alpha$. We conclude the proof by noting that the convergence rate $\mathcal{O}(\rho(\nabla_{\mathbf{x}} G_{\text{LA-EG}_{\text{Sim}}})^{\frac{t}{k}})$ of $G_{\text{LA-EG}_{\text{Sim}}}$ provided by Lemma 15 is faster than the rate $\mathcal{O}(\rho(\mathbf{M})^t)$ of $F_{\text{EG}_{\text{Sim}}}$, assuming amortization of computations over $k$ forward steps. $\square$

### E.9 PROOF OF THEOREM 9

*Proof.* From Equation 7, the Jacobian of $G$ evaluated at $\mathbf{x}^*$ can written as

$$\nabla_{\mathbf{x}} G(\mathbf{x}^*) = \nabla_{\mathbf{x}} \left( (1-\alpha)\text{id} + \alpha F^k \right) (\mathbf{x}^*) = (1-\alpha)\mathbf{I} + \alpha \nabla_{\mathbf{x}} F^k(\mathbf{x}^*) \tag{52}$$

$$= (1-\alpha)\mathbf{I} + \alpha \prod_{i=1}^{k} \nabla_{\mathbf{x}} F(F^{i-1}(\mathbf{x}^*)) = (1-\alpha)\mathbf{I} + \alpha(\nabla_{\mathbf{x}} F(\mathbf{x}^*))^k, \tag{53}$$

where the chain rule is used in third equality with a slight abuse of notation $F^0 \stackrel{\text{def}}{=} \text{id}$. We use the fact that $\mathbf{x}^*$ is an equilibrium of dynamics $F$ for the last equality. It is easy to see from Equation 53 that eigenvalues of $\nabla_{\mathbf{x}} G(\mathbf{x}^*)$ can be written as $1 - \alpha + \alpha \lambda_i^k$ for each $\lambda_i \in \lambda(\nabla_{\mathbf{x}} F(\mathbf{x}^*))$.

However, $\lambda_i^k$ is either on/inside $S_1$ since $|\lambda_i| \le 1$ for each $i$ due to the Lyapunov stability of $\mathbf{x}^*$ in $F$. Therefore, $1 - \alpha + \alpha \lambda_i^k$ is an interpolation between two points $(1, 0) \in S_1$ and $\lambda_i^k$ either on/inside $S_1$; hence $|1 - \alpha + \alpha \lambda_i^k| \le 1$. By assumption that $\lambda_i^k \ne (1, 0)$ for each $\lambda_i \in \lambda(\nabla_{\mathbf{x}} F(\mathbf{x}^*))$, the inequality is strict, *i.e.* $|1 - \alpha + \alpha \lambda_i^k| < 1$, implying the local asymptotic stability of $\mathbf{x}^*$ in $G$ by Proposition 14. $\qquad\square$

### E.10    Proof of Theorem 10

*Proof.* From Equation 7, the Jacobian of $G$ evaluated at $\mathbf{x}^*$ can written as

$$\nabla_{\mathbf{x}} G(\mathbf{x}^*) = \nabla_{\mathbf{x}} \left( (1-\alpha)\text{id} + \alpha F^k \right)(\mathbf{x}^*) = (1-\alpha)\mathbf{I} + \alpha \nabla_{\mathbf{x}} F^k(\mathbf{x}^*) \tag{54}$$

$$= (1-\alpha)\mathbf{I} + \alpha \prod_{i=1}^{k} \nabla_{\mathbf{x}} F(F^{i-1}(\mathbf{x}^*)) = (1-\alpha)\mathbf{I} + \alpha(\nabla_{\mathbf{x}} F(\mathbf{x}^*))^k, \tag{55}$$

where the chain rule is used in third equality with a slight abuse of notation $F^0 \stackrel{\text{def}}{=} \text{id}$. We use the fact that $\mathbf{x}^*$ is an equilibrium of dynamics $F$ for the last equality. It is easy to see from Equation 55 that the eigenvalues of $\nabla_{\mathbf{x}} G(\mathbf{x}^*)$ can be written as $1 - \alpha + \alpha \lambda_i^k$ for each $\lambda_i \in \lambda(\nabla_{\mathbf{x}} F(\mathbf{x}^*))$.

Now assume that every element of $\lambda_{\ge 1}(\nabla_{\mathbf{x}} F(\mathbf{x}^*))$ has non-zero imaginary part, and let $k \in \left( \frac{\pi}{2\theta_{\min}(\lambda_{\ge 1}(\nabla_{\mathbf{x}} F(\mathbf{x}^*)))}, \frac{3\pi}{2\theta_{\max}(\lambda_{\ge 1}(\nabla_{\mathbf{x}} F(\mathbf{x}^*)))} \right)$. Let $\eta > 0, \mathbf{J} \in \mathbb{R}^{n \times n}$ be such that $\nabla_{\mathbf{x}} F(\mathbf{x}^*) = \mathbf{I} - \eta \mathbf{J}$. Then by Lemma 4, $\Re(\lambda_i^k) < 0$ holds for any $\lambda_i \in \lambda_{\ge 1}(\nabla_{\mathbf{x}} F(\mathbf{x}^*))$, and the existence of such $k \in \mathbb{N}$ is guaranteed for a small enough $\eta$ when $\frac{\Im_{\max}(\lambda_{\ge 1}(\nabla_{\mathbf{x}} F(\mathbf{x}^*)))}{\Im_{\min}(\lambda_{\ge 1}(\nabla_{\mathbf{x}} F(\mathbf{x}^*)))} < 3$. Then it follows from the second case of Theorem 3 that $\rho(\nabla_{\mathbf{x}} G(\mathbf{x}^*)) < 1$ holds for a small enough $\alpha$. By Proposition 14, this implies local asymptotic stability of $\mathbf{x}^*$ in $G$, concluding the proof. $\qquad\square$

### E.11    Proof of Theorem 11

*Proof.* From Equation 7, the Jacobian of $G$ evaluated at $\mathbf{x}^*$ can written as

$$\nabla_{\mathbf{x}} G(\mathbf{x}^*) = \nabla_{\mathbf{x}} \left( (1-\alpha)\text{id} + \alpha F^k \right)(\mathbf{x}^*) = (1-\alpha)\mathbf{I} + \alpha \nabla_{\mathbf{x}} F^k(\mathbf{x}^*) \tag{56}$$

$$= (1-\alpha)\mathbf{I} + \alpha \prod_{i=1}^{k} \nabla_{\mathbf{x}} F(F^{i-1}(\mathbf{x}^*)) = (1-\alpha)\mathbf{I} + \alpha(\nabla_{\mathbf{x}} F(\mathbf{x}^*))^k, \tag{57}$$

where the chain rule is used in third equality with a slight abuse of notation $F^0 \stackrel{\text{def}}{=} \text{id}$. We use the fact that $\mathbf{x}^*$ is an equilibrium of dynamics $F$ for the last equality. It is easy to see from Equation 57 that the eigenvalues of $\nabla_{\mathbf{x}} G(\mathbf{x}^*)$ can be written as $1 - \alpha + \alpha \lambda_i^k$ for each $\lambda_i \in \lambda(\nabla_{\mathbf{x}} F(\mathbf{x}^*))$.

Now assume that every element of $\lambda_{\max}(\nabla_{\mathbf{x}} F(\mathbf{x}^*))$ has non-zero imaginary part, and let $k \in \left( \frac{\pi}{2\theta_{\min}(\lambda_{\max}(\nabla_{\mathbf{x}} F(\mathbf{x}^*)))}, \frac{3\pi}{2\theta_{\max}(\lambda_{\max}(\nabla_{\mathbf{x}} F(\mathbf{x}^*)))} \right)$. Let $\eta > 0, \mathbf{J} \in \mathbb{R}^{n \times n}$ be such that $\nabla_{\mathbf{x}} F(\mathbf{x}^*) = \mathbf{I} - \eta \mathbf{J}$. Then by Lemma 4, $\Re(\lambda_i^k) < 0$ holds for any $\lambda_i \in \lambda_{\max}(\nabla_{\mathbf{x}} F(\mathbf{x}^*))$, and the existence of such $k \in \mathbb{N}$ is guaranteed for a small enough $\eta$ when $\frac{\Im_{\max}(\lambda_{\max}(\nabla_{\mathbf{x}} F(\mathbf{x}^*)))}{\Im_{\min}(\lambda_{\max}(\nabla_{\mathbf{x}} F(\mathbf{x}^*)))} < 3$. Then it follows from the third case of Theorem 3 that $\rho(\nabla_{\mathbf{x}} G(\mathbf{x}^*)) < \rho(\nabla_{\mathbf{x}} F(\mathbf{x}^*))^k$ holds for a small enough $\alpha$. We conclude the proof by noting that this implies the upper bound $\mathcal{O}(\rho(\nabla_{\mathbf{x}} G(\mathbf{x}^*)^{\frac{t}{k}})$ on the rate of local convergence provided by Proposition 14 is faster than $\mathcal{O}(\nabla_{\mathbf{x}} F(\mathbf{x}^*)^t)$. $\qquad\square$

### E.12    Proof of Proposition 12

*Proof.* We directly follow the proofs of Lemma 2.1 and Lemma 3.1 in Daskalakis & Panageas (2018) and show that $\alpha \in (0, \frac{1}{1+L^k})$ guarantees locally diffeomorphic Lookahead dynamics, *i.e.*, it is locally invertible at any given points.

Note from Equation 7 that the Jacobian of $G$ evaluated at $\mathbf{x}$ can written as

$$\nabla_{\mathbf{x}} G(\mathbf{x}) = \nabla_{\mathbf{x}} \left( (1-\alpha)\mathrm{id} + \alpha F^k \right)(\mathbf{x}) = (1-\alpha)\mathbf{I} + \alpha \nabla_{\mathbf{x}} F^k(\mathbf{x}) \tag{58}$$

$$= (1-\alpha)\mathbf{I} + \alpha \prod_{i=1}^{k} \nabla_{\mathbf{x}} F(F^{i-1}(\mathbf{x})), \tag{59}$$

where we have used the chain rule in the last equality with a slight abuse of notation $F^0 \stackrel{\text{def}}{=} \mathrm{id}$.

Now assume that $\alpha \in \left( 0, \frac{1}{1+L^k} \right)$ and consider the following inequalities

$$\rho(\prod_{i=1}^{k} \nabla_{\mathbf{x}} F(F^{i-1}(\mathbf{x}))) \le \left\| \prod_{i=1}^{k} \nabla_{\mathbf{x}} F(F^{i-1}(\mathbf{x})) \right\| \le \prod_{i=1}^{k} \left\| \nabla_{\mathbf{x}} F(F^{i-1}(\mathbf{x})) \right\| \le L^k, \tag{60}$$

where the first and second inequalities hold for any operator norms and the last inequality is due to $L$-Lipschitzness of $F$. Then it follows from the assumption that

$$\rho(\prod_{i=1}^{k} \nabla_{\mathbf{x}} F(F^{i-1}(\mathbf{x}^*))) \le L^k < \frac{1-\alpha}{\alpha}. \tag{61}$$

Therefore, we conclude that $G$ locally diffeomorphic, since $\rho(\prod_{i=1}^{k} \nabla_{\mathbf{x}} F(F^{i-1}(\mathbf{x}^*))) < \frac{1-\alpha}{\alpha}$ implies $0 \notin \lambda(\nabla_{\mathbf{x}} G(\mathbf{x}))$.

Now let us define the set of unstable equilibria of $G$ as $U \stackrel{\text{def}}{=} \{\mathbf{x}^* : G(\mathbf{x}^*) = \mathbf{x}^*, \rho(\nabla_{\mathbf{x}} G(\mathbf{x}^*)) > 1\}$. Then it directly follows from the locally diffeomorphic $G$ and the arguments of Lee et al. (2019); Daskalakis & Panageas (2018) that the set $\{x^{(0)} : \lim_{t \to \infty} G^t(x^{(0)}) \in U\}$ is of measure zero, which concludes the proof. We refer the readers to Appendix A of Daskalakis et al. (2018) for the detailed derivation of measure-zero arguments. $\qquad\square$

### E.13 PROOF OF PROPOSITION 13

*Proof.* From Equation 7, the Jacobian of $G$ evaluated at $\mathbf{x}^*$ can written as

$$\nabla_{\mathbf{x}} G(\mathbf{x}^*) = \nabla_{\mathbf{x}} \left( (1-\alpha)\mathrm{id} + \alpha F^k \right)(\mathbf{x}^*) = (1-\alpha)\mathbf{I} + \alpha \nabla_{\mathbf{x}} F^k(\mathbf{x}^*) \tag{62}$$

$$= (1-\alpha)\mathbf{I} + \alpha \prod_{i=1}^{k} \nabla_{\mathbf{x}} F(F^{i-1}(\mathbf{x}^*)) = (1-\alpha)\mathbf{I} + \alpha(\nabla_{\mathbf{x}} F(\mathbf{x}^*))^k, \tag{63}$$

where the chain rule is used in third equality with a slight abuse of notation $F^0 \stackrel{\text{def}}{=} \mathrm{id}$. We use the fact that $\mathbf{x}^*$ is an equilibrium of dynamics $F$ for the last equality. It is easy to see from Equation 63 that the eigenvalues of $\nabla_{\mathbf{x}} G(\mathbf{x}^*)$ can be written as $1 - \alpha + \alpha \lambda_i^k$ for each $\lambda_i \in \lambda(\nabla_{\mathbf{x}} F(\mathbf{x}^*))$. However, by the assumption, there exists a $\lambda \in \lambda(\nabla_{\mathbf{x}} F(\mathbf{x}^*))$ such that $|\lambda| > 1$. Since $\lambda$ is a positive real number, we have $|1 - \alpha + \alpha \lambda^k| > 1$, concluding the proof. $\qquad\square$

## F ADDITIONAL EXPERIMENTS

### F.1 EIGENVALUES OF GAN DYNAMICS

Theorem 10-11 assumes the *radius-supporting eigenvalues*, namely $\lambda_{\ge 1}(\nabla_{\mathbf{x}} F)$ and $\lambda_{\max}(\nabla_{\mathbf{x}} F)$), to have non-zero imaginary parts and imaginary conditioning less than 3; otherwise, the existence of $k$ that satisfies the sufficient conditions of Theorem 10-11 may not exist. We verify whether such assumptions are realistic in practical settings. Specifically, we train GANs on MNIST dataset with two different loss functions, non-saturating (Goodfellow et al., 2014) and WGAN-GP (Gulrajani et al., 2017), and visualize the top 20 eigenvalues of $\nabla_{\mathbf{x}} F_{\mathrm{GD_{Sim}}}$ for each loss function in Figure 5.

Figure 5 suggests most of the radius-supporting eigenvalues of $\nabla_{\mathbf{x}} F_{\mathrm{GD_{Sim}}}$ at well-performing point (Inception Score (IS) (Salimans et al., 2016) $\approx 9$) are distributed along the imaginary axis, and have non-zero imaginary part with imaginary conditioning less than 3. This suggests that our assumptions on the eigenvalues is not unrealistic and Theorem 10-11 can be applied for a practical non-linear game like GANs.

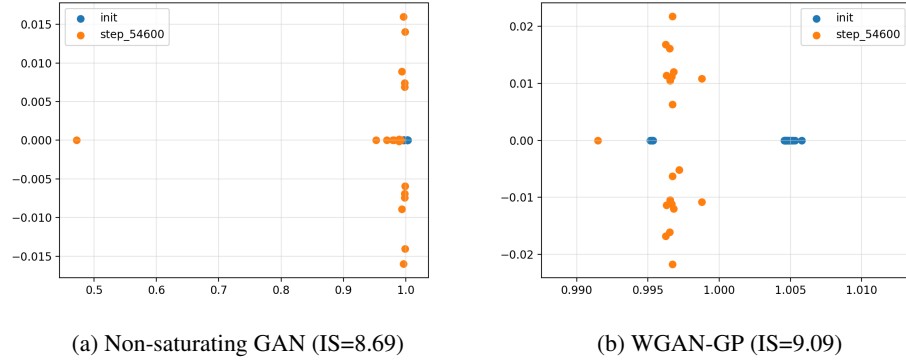

(a) Non-saturating GAN (IS=8.69)    (b) WGAN-GP (IS=9.09)

Figure 5: Visualized top 20 eigenvalues of $\nabla_{\mathbf{x}} F_{\text{GD}_{\text{Sim}}}$ before (blue) and after (orange) training GANs with two different loss functions on MNIST.

## F.2 ILL-CONDITIONED BILINEAR GAMES AND MOMENTUM METHODS

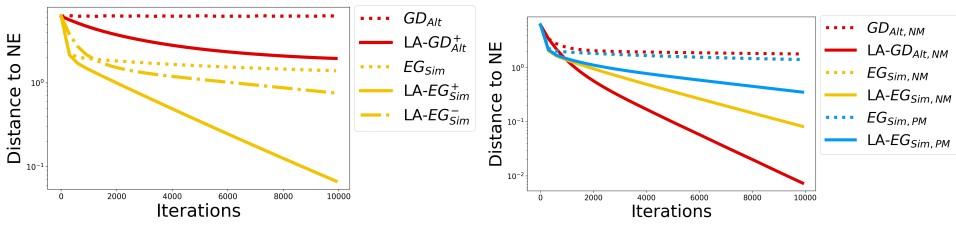

(a) Lookahead dynamics without momentum.    (b) Lookahead dynamics with momentum.

Figure 6: Optimization progress of first-order methods in an ill-conditioned bilinear game. **(a)** Comparison between convergence of Lookahead dynamics chosen by $(+)$ and $(-)$ against Theorem 5 and 8. **(b)** Comparison between convergence of Lookahead dynamics with positive (PM) and negative (NM) momentums.

We test the convergence and acceleration of Lookahead dynamics in an ill-conditioned bilinear game, and see if Lookahead can accelerate momentum-based dynamics in such game. Specifically, we test convergence of each dynamics in the game given by Equation 12 with $n = 20$ and $\epsilon = 1$, which gives a sample of $\mathbf{A}$ with $\sigma_{\max} = 8.81$ and $\sigma_{\min} = 0.11$. Note that this game has a significantly larger conditioning number $\frac{\sigma_{\max}}{\sigma_{\min}} = 76.4$ than the bilinear game of a conditioning 1.401 we used in Section 5.

We fix $\eta = 0.05$ throughout the experiments, and use Theorem 8 to derive theoretically recommended $(+)$ hyperparameters $k = 300, \alpha = 0.9$ for LA-EG$_{\text{Sim}}$ dynamics. We use $k = 50, \alpha = 0.1$ to represent hyperparameters of LA-EG$_{\text{Sim}}$ chosen against $(-)$ the theorem. For LA-GD$_{\text{Alt}}$, we use $k = 300, \alpha = 0.1$. We use the momentum factor $\beta = -0.1$ for negative (NM) and $\beta = 0.1$ for positive (PM) momentum methods.

Figure 6 (a) shows that Theorem 5 and 8 indeed hold even for an ill conditioned game. Furthermore, Figure 6 (b) suggests that Lookahead can significantly accelerate the convergence of momentum methods that provably perform well on bilinear games, including the gradient descent with negative momentum (Gidel et al., 2019b) and extragradient with momentum (Azizian et al., 2020).

