# OpenReview forum: "Characterizing Lookahead Dynamics of Smooth Games"
_ICLR.cc/2021/Conference — Reject_

### Official Review · AnonReviewer3 · 2020-10-22
**An important problem to study but the results are somewhat limited and not so well presented**

**Rating:** 7
**Confidence:** 4

**Review:**

### Summary

Motivated by the empirical success of lookahead in GAN training, the paper studies theoretically the convergence behavior of the algorithm in smooth games. In particular, the authors focus on bilinear games and local convergence around equilibrium points and derive sufficient conditions under which lookahead improves upon its base dynamics (by either stabilizing a non-convergent algorithm or accelerating a convergent one).

### Pros

The lookahead optimizer studied in this paper is relatively new and worths more theoretical investigation.  In this spirit, this work consists in, as far as I am aware, the first attempt to understand theoretically the performance of lookahead in game optimization. The results concerning the potential stabilization and acceleration achieved by the lookahead mechanics provide us some insight into the reasons for its empirical success.

### Cons

#### On the significance of the results
1. The sufficient conditions provided by the authors appear to be too restricted and cannot paint a global picture of what would happen in practice (even in the bilinear case).
In more than half of the theorems, we require the (imaginary) condition number to be smaller than 3. Is this realistic? What happens if this is not verified? Is it even reasonable to suppose that every eigenvalue of the matrix $J$ has non-zero imaginary part given that we would be at the other opposite if we consider a minimization problem (every eigenvalue would be real)?
The work would be much more complete if the authors could also discuss what may happen if these sufficient conditions are not verified.
2. The implication of Theorem 12 is unclear. The fact that the dynamics avoids unstable equilibrium points of the dynamics itself seems to be of little interest for the problem that we are solving (and is a quite standard result). In effect, we are more interested in the characterization of the set of stable equilibrium points which may contain undesired solutions. This is clearly illustrated by the nonlinear game experiment in which the algorithm converges to a spurious attractor of the problem (to be explained in more detail below).

#### On the presentation of the results
The presentation of the results needs to be improved. In its current form, the authors provide plenty of theorems that could be difficult to decipher for the readers.
1. For example, it seems that both Theorems 3 and 4 are mainly used for deriving the rest of the theorems. This can be explicitly stated and I even feel it would be better to call them lemmas instead of theorems since it is not immediately clear what these two results imply.
2. The authors could comment on the existence of $k$ for Theorems 5~11. While this is easy to derive it would be helpful to clearly state this somewhere (such as a sentence saying that such $k$ will always exist in all the theorems.)
3. For both Theorems 10 and 11, corollaries for different base dynamics under different assumptions on the Jacobian of the vector field can be provided. Again, this may be straightforward but can help to understand the results.

### Detailed comments

#### Bilinear game reduction

For the paragraphs at the bottom of page 5. The authors describe why we only need to be interested in a bilinear game of the form (11). I understand this is due to the fact that modulo some transformation (9) and (11) have in fact the same trajectory. However, given the presentation of the paper, we may feel that to solve (9) we first solve (11) but this does not really make sense because the real goal is not to solve the problem (otherwise for (11) we know the solution is the all-zero vectors) but to understand how the algorithms perform in this problem. I feel that this point is much better explained in, for example, Zhang & Yu (2020).

#### Diagonalizability

I fail to understand the authors' arguments about the diagonalizability of the matrices in the proofs of Theorems 5\~8.  Nonetheless, I feel this can be easily proved by using the fact that all real symmetric and skew-symmetric matrices are diagonalizable and we do not even need the extra assumptions on $k$ of Theorems 6~8.

#### Nonlinear game experiment

In my opinion, the nonlinear game example considered in the experiments is not very appropriate. Actually, it depends on the problem that we are looking at. For (13), the origin is not a saddle point and not even a min-max solution (while it is a max-min solution). If our goal is to find the zeros of the vector field, we indeed want a convergence to the point. However, the whole paper is motivated by the computation of a (local) nash equilibrium, then we would like to avoid this point. Notice how this example is presented in (Hsieh et al. 2020): we want to escape from the origin and not converge to it.
While the above points are not in conflict with what the authors want to demonstrate, this should be made clear to the readers to avoid confusion.

#### A very minor point
Equation (25) and the following analysis are actually for the proximal method, not EG.

---

> ### Author Response · Authors · 2020-11-19
> **Answers to Reviewer3 (1/2)**
>
> We sincerely thank Reviewer3 for constructive and insightful comments.
>
> In the newly uploaded version of the paper, we have largely improved the theoretical significance of our results and the quality of their presentation. We believe that this addresses most concerns of Reviewer3. Please see blue fonts in Section 3-4 in the newly updated draft to check how our paper is updated.
>
> Below, we address each comment of Reviewer3 in detail.
>
> # 1. Restrictive assumptions on eigenvalues
>
> We acknowledge that in the initial draft, the assumptions on the eigenvalues were fairly strong, hence restrictive in the sense that they require the *entire* eigenvalues to have non-zero imaginary parts and their imaginary conditioning to be less than 3.
>
> In the updated version of the paper, we significantly relaxed these assumptions (see Lemma 3-4 in the updated draft) and now we require the same conditions to hold only for a *subset* of the eigenvalues, namely $\lambda_{1\geq}(\textbf{X})$ and $\lambda_{max}(\textbf{X})$, that we call *radius-supporting eigenvalues*. In Appendix F.1, we visualize eigenvalues of GANs trained on MNIST dataset, and demonstrate that these relaxed assumptions are realistic even for a practical nonlinear game like GANs.
>
> As a consequence, Theorem 7-8 (Acceleration of PP and EG) are now applicable to any general bilinear games, including the ill-conditioned ones. Furthermore, Theorem 10-11 (Local stabilization and acceleration) are now also applicable to more general settings. For example, the updated version of Theorem 10-11 even covers the case where some eigenvalues lie on the real axis (as long as they do not belong to radius-supporting eigenvalues), and can be applied for any equilibrium with well-conditioned (imaginary conditioning less than 3) radius-supporting eigenvalues.
>
>
>
> # 2.  Characterization of stable equilibria
>
> First, we would like to clarify that Theorem 9-10 of the initial draft are already (implicitly) characterizing the stable equilibria of Lookahead dynamics: Theorem 9 shows that any LASE of base dynamics is also a LASE of Lookahead dynamics, and Theorem 10 implies that such inclusion is strict in general.
>
> In the updated version of the paper, we extend Theorem 9 a bit and show that any Lyapunov stable equilibrium (SE) of a dynamics is a LASE of its Lookahead dynamics; hence SE $\subset$ Lookahead-LASE $\subset$ Lookahead-SE, where each inclusion is strict in general.
>
> In fact, such strict inclusion is a double-edged sword in the context of Nash equilibrium (NE) computation: while it could be helpful if Lookahead stabilizes unstable NE (e.g., bilinear games), it also carries a possibility for introducing non-Nash LASE (e.g., the nonlinear game in Section 5), which would be a bad thing for computing NE. Therefore, the stabilization effect of Lookahead is quite double-sided, and its overall impact on NE computation depends on the global structure of the game and base dynamics.
>
> To make such double-sided implications explicit to the readers, we added a paragraph in Section 4 to discuss the consequence of Theorem 9-10 in the context of NE computation.
>
>
> # 3. Relevance of Theorem 12
>
> We admit that the discussion on the implication of Theorem 12 (Proposition 12 in the updated draft) was quite cursory. To provide a better understanding for the actual consequence of Theorem 12, we added Proposition 13, which states that Lookahead dynamics preserves unstable equilibria in fully-cooperative games (e.g., minimization problems).
>
> Together with Theorem 12, Proposition 13 guarantees that the randomly-initialized iterates of Lookahead dynamics can almost surely avoid unstable points of the base dynamics (e.g., local maxima) in fully-cooperative games. This assures the sanity of Lookahead optimizer in minimization problems. Note that such result is in contrast with the case of non-fully-cooperative games (i.e., games with non-real eigenvalues) where Lookahead might stabilize (possibly undesirable) unstable points (Theorem 10), as pointed by Reviewer3.

---

> ### Author Response · Authors · 2020-11-19
> **Answers to Reviewer3 (2/2)**
>
> # 4. Presentation of the results
>
> We thank Reviewer3 for suggesting the detailed points to improve the presentation of our results. In the newly uploaded version of the paper, we updated the following points to reflect the suggestions of Reviewer3:
>
> - We cleaned up the overall statements of Theorem 5-8 and Theorem 10-11.
> - We renamed Theorem 3-4 to Lemma 3-4, and added a sentence in Section 3.1 that explicitly mentions their usage throughout the paper.
> - We factored out the statements on the existence of $k$ (Theorem 5-8, Theorem 10-11) to the main paragraphs, and explicitly mention that they may not exist in certain cases.
>
> While we agree with Reviewer3 that corollaries of Theorem 10-11 will help readers to better understand the results, we decided not to include them in the updated draft since it is almost impossible to add 8 extra corollaries (= 2 (Theorem 10-11) x 4 (GD_{Alt/Sim}, EG_{Sim}, PP_{Sim})) to the paper, considering the page limit.
>
>
> # 5. Misleading description of the bilinear game reduction
>
> We thank Reviewer3 for pointing out that the paragraph in Section 3.2 that justifies the bilinear game reduction can be quite misleading. In the newly uploaded version, we replaced *“... we can solve Equation 10 by solving a rather simpler problem … and then changing the coordinates ...”* with *“... we can analyze the dynamics of Equation 10 by inspecting a rather simpler problem … as they are equivalent up to some rotations and translations.”*.
>
>
> # 6. Necessity of matrix diagonalizability
>
> We greatly thank Reviewer3 for pointing out on the necessity of diagonalizability arguments in our proofs. Such arguments were mainly due to the first case of Lemma 15, which we use as a main tool for proving Theorem 5-8.
>
> However, as Reviewer3 pointed out, any linear system with the spectral radius less than 1 is exponentially stable and there is no need for the transfer matrix to be diagonalizable at all.
>
> In the newly uploaded version, we removed the diagonalizability arguments in the proofs of Theorem 5-8, and thus Theorem 6-8 now does not require modulo conditions on $k$. Note, however, that we still need the modulo condition in Theorem 5 to prevent the case $\lambda^k=(1,0)$.
>
>
> # 7. Clarification on the nonlinear game experiment
>
> Reviewer3 pointed out that the nonlinear game experiment in Section 5 is not appropriate since the equilibrium (0, 0) of the game is not a Nash equilibrium.
>
> However, we would like to emphasize that the goal of our experiments is to empirically verify our theoretical predictions, and we did not claim that Lookahead ‘only’ converges to a (local) Nash equilibrium in general (nonlinear) games. In fact, the results of nonlinear game experiments are exactly what our theories promise: local stabilization and acceleration to a given equilibrium.
>
> We understand Reviewer3’s point that it would be better to mention that (0, 0) is not a Nash equilibrium of the nonlinear game. We will make this point clear in the final draft and make sure to clarify the implications of our experiments.

---

> > ### Comment · AnonReviewer3 · 2020-11-19
> > **The updated version has both results and presentation greatly improved; score increased from 5 to 7**
> >
> > I appreciate the efforts the authors have made to address the reviewers' comments. I find the updated version much more readable and the implications of the theorems are now clear. I increased my score from 5 to 7 (indeed hesitating between 6 and 7) as most of my concerns have been addressed (though the new condition can still be hard to decipher).
> > As pointed out by reviewer 2, it will be interesting to study (experimentally) what happens when the sufficient conditions of several theorems are not verified, but I think the paper as updated is already ready for publication (I did not check the updated proofs). Note that what I mentioned as "a very minor point" has not been addressed and should probably be corrected as well.

---

> > > ### Author Response · Authors · 2020-11-24
> > > **Thank you for re-evaluating our work**
> > >
> > > We deeply appreciate Reviewer3 for taking the time to re-evaluate our updated draft.
> > >
> > > We agree that further analysis on the non-necessity of our theorems would be an interesting future work. “a very minor point” on the derivation of hyperparameters for EG in Appendix D, is now fixed; EG experiments in Section 5 are correspondingly updated, and the results are the same.

---

### Official Review · AnonReviewer4 · 2020-10-27
**Report**

**Rating:** 9
**Confidence:** 4

**Review:**

Summary: This paper investigates „lookahead dynamics of smooth games“. By this the authors mean discrete-time dynamical systems generating from a given algorithm by adding a relaxation step in the updates. The main aim of the paper is to solve smooth games. Under sufficient convexity assumptions Nash equilibria for such games can be identified as solutions to a Variational Inequality with a monotone and operator. This is in particular the case for convex-concave min-max problems. The main conclusion of this paper is that a combination of relaxation and lookahead effects stabilizes the learning dynamics and can lead to acceleration over the base algorithm.

Evaluation: This is a very strong paper with an extremely large number of interesting results. In my opinion it makes an extremely good contribution to the flourishing literature  on game dynamics. I only have some small technical remarks which can easily be fixed.


Specific Remarks:

 .  Interchange the order of eqs. (5) and (6).
. Define $F^{k}$ in eq. (7)
. Check for consistency of notation: Sometimes $M_{m\times m}$ is used for the matrix space, then $\mathbb{R}^{m\times m}$. If the former is used, explain which field of numbers is used.
. Define $\rho$ in Theorem 3

---

> ### Author Response · Authors · 2020-11-19
> **Answers to Reviewer4**
>
> We thank Reviewer4 for exceptionally positive comments. We addressed the concern on Eq. 5-6 by adding ‘where,’ to Eq. 5. We also updated some notations in the new version of the paper. Please see blue fonts in the updated draft to check how our paper is updated.

---

### Official Review · AnonReviewer2 · 2020-10-29
**An interesting idea that could be developed further**

**Rating:** 4
**Confidence:** 4

**Review:**

## Summary
This paper provides a spectral analysis of Lookahead dynamics. This paper’s main results state that lookahead dynamics can improve a given method’s convergence rate if the Jacobian’s operator has eigenvalues with non-zero imaginary parts and an imaginary conditioning (ration between the largest and the smallest imaginary part) smaller than 3.

Overall I think that the idea of this paper is exciting and is motivated by recent empirical observation. However, the theoretical results of this paper are relatively weak because of some restrictive assumptions.

### Pros
The motivations of this paper are clean and propose a promising line of analysis.

### Cons
The theory is only developed in a very restrictive setting (Bilinear with condition number smaller than 3 and general games with non-real eigenvalues and imaginary condition number smaller than 3) where the optimization problem is easy to solve. (a small condition number corresponds to an easy optimization problem.)
The experiments are relatively weak since they are only on well-conditioned bilinear games, a 2D non-linear problem, and do not explore the necessity of the assumptions of the theorems of the paper.

## Questions/comment:

- It seems to me that the condition that the Jacobian has non-real eigenvalue and $\frac{\mathcal I_\max}{\mathcal I_\min} < 3$ is an artifact from your technique proof because lookahead does converge in the context of minimization (only real eigenvalues).
Though in the minimization case, it seems quite direct to see that look ahead slows down the dynamics $ 1 - \alpha + \alpha \lambda > \lambda \, \forall 1>\lambda >0$ the interesting property of lookahead highlighted by this work is that it can positively affect the impact eigenvalues with a large imaginary part on the convergence.
Since in games, it has been shown that eigenvalues with a large imaginary part may slow down the convergence of games [Mescheder 2017], lookahead seems to be a promising direction to improve the convergence rate of the gradient method. Specifically, in the context of games with large or infinite imaginary condition number.

- Overall, I think that this paper would benefit from a more precise analysis of the convergence rates.  Then answer the following question: can we find some problems where lookahead provides a significant improvement in terms of convergence rate (against standard methods such as Extragradient or Gradient).
The concept of acceleration usually refers to a significant improvement of the convergence rate (see for instance [Azizian et al. 2020] for a discussion on acceleration on games.) In the case of Theorem 3 it would mean $\rho(f(X)) << \rho(X)^k$.


- Also, I am not sure about the relevance of the “local stabilization properties”, usually one may want to diverge from certain points where $\rho(\nabla_x F(x)) >1$ e.g., local maxima.
What can you say regarding the fact that Lookahead does not converge to “bad” stationary points (for instance, in multi-objective minimization one want to avoid a local maximum for each player’s loss)?

- Regarding your experiment, you could try to test if the conditions in your theorem are necessary:
Can I find hyperparameters to make lookahead converging even when there are real eigenvalues or a condition number larger than 3?
Can lookahead provide improvement with respect to optimization methods that provably perform well on bilinear minimax such as extragradient with momentum or specific method for bilinear [Azizian et al. 2020]?


Azizian, Waïss, et al. "Accelerating smooth games by manipulating spectral shapes." AISTATS (2020).


Mescheder, Lars, Sebastian Nowozin, and Andreas Geiger. "The numerics of gans." Advances in Neural Information Processing Systems. 2017.

---

> ### Author Response · Authors · 2020-11-19
> **Answers to Reviewer2**
>
> We greatly thank Reviewer2 for thoughtful and constructive comments.
>
> In the newly uploaded version of our paper, we largely improved the theoretical significance of our results by relaxing the assumptions on the eigenvalues. We also verify that the relaxed assumptions can hold in a practical nonlinear game like GANs. Below, we address each comment in detail. Please see blue fonts in the newly updated draft to check how our paper is updated.
>
> # 1. Restrictive assumptions on the spectrum of Jacobian
>
> We admit that the assumptions on the spectrum of base dynamics in the initial draft were very strong, and therefore our theoretical results were limited to restricted settings.
>
> In the updated draft, we significantly relax these assumptions (see Lemma 3-4 in Section 3.1). The initial assumptions required the *entire* eigenvalues to have non-zero imaginary parts and imaginary conditioning number less than 3. On the other hand, the relaxed version now requires the same condition to hold only for a *subset* of the eigenvalues, namely $\lambda_{1\geq}(\textbf{X})$ and $\lambda_{max}(\textbf{X})$.
>
> With the relaxed assumptions, we also updated Theorem 5-8 and Theorem 10-11. Now Theorem 7-8 (Acceleration of PP, EG) can be applied even for the ill-conditioned bilinear games, and this is verified by the experiment in Appendix F.2. Furthermore, Theorem 10-11 now covers the case where some real eigenvalues coexist with non-real eigenvalues (under assumption that the real eigenvalues does not belong to $\lambda_{1\geq}(\nabla_{\textbf{x}} F)$ and $\lambda_{max}(\nabla_{\textbf{x}} F))$. Experiments in Appendix F.1 (added in the updated draft)  verify that such assumptions can hold even for a very practical nonlinear game like GANs.
>
>
> # 2. Precise analysis of convergence rate
>
> We agree with Reviewer2 that our work would be more complete if we could provide optimal $k$ and $\alpha$, and analyze the bounds of acceleration that could be made upon a base dynamics.
>
> While the experiments in Section 5 and Appendix F.2 (added in the updated draft) actually show that Lookahead can indeed significantly accelerate EG dynamics, it is a mathematically non-trivial problem to analyze the (optimal) spectral radius of Lookahead dynamics $\min_{k\in\mathbb{N}, \alpha\in(0,1)} \max_{\lambda_i} |1-\alpha+\alpha\lambda_i^k|^{1/k}$, especially considering the fact that each $\lambda_i^k$ has a periodic nature.
>
> We believe this is indeed an important problem to study as an intriguing future work.
>
>
> # 3. Relevance of the local stabilization property
>
> Reviewer2 pointed out that one may usually want to diverge from unstable points, hence Theorem 10 could be something that we do not want in certain settings (e.g., minimization problems).
>
> First, we would like to clarify that Theorem 10 requires non-real eigenvalues, and therefore does not apply to fully-cooperative games (e.g., minimization problems) that Reviewer2 mentioned; such games exhibit real-eigenvalues only. In fact, Lookahead preserves instability in fully-cooperative games: $|1-\alpha + \alpha\lambda^k| > 1$ if $\lambda > 1$. Proposition 12 in the updated draft (Previously Theorem 12) guarantees that Lookahead can avoid such points. Therefore, we can actually guarantee that Lookahead almost surely avoids local maxima in minimization problems. To make this point clear, we add Proposition 13 and show that Lookahead preserves instability of fully cooperative games.
>
>
> # 4. Necessity of the sufficient conditions
>
> The sufficient conditions on $k$ and $\alpha$ in Theorem 6-8 and 10-11 are not necessary. Actually, this is clear from Lemma 3-4 in the updated draft, the main tools that we use for our proofs.
>
> While Lemma 3 requires $\Re(\lambda^k)<1$ and $\Re(\lambda^k)<\rho(\textbf{X})^{2k}$ to reduce the spectral radius of $\textbf{X}$, Lemma 4 provides conditions to guarantee $\Re(\lambda^k) < 0$, which is not necessary for $\Re(\lambda^k)<1$ and $\Re(\lambda^k)<\rho(\textbf{X})^{2k}$. Therefore, the sufficient conditions in Theorem 6-8 and 10-11, which are built upon Lemma 3-4, are not necessary as well.
>
> A concrete consequence of this non-necessity can be observed in the experiment of Appendix F.2. Figure 6 (a) shows that LA-EG dynamics with $k$ and $\alpha$ that does not satisfy Theorem 8, denoted by (-), accelerates EG dynamics.
>
> # 5. Additional experiments on (ill-conditioned) bilinear games
>
> To see if (1) the convergence and acceleration guarantees (Theorem 5 and 8) hold even for an ill-conditioned bilinear game, and (2) test if Lookahead can improve advanced momentum methods that Reviewer2 mentioned, we conducted additional experiments in Appendix F.2. Figure 6. (a) suggests that our updated theorems actually hold even for ill-conditioned games. Furthermore, Figure 6. (b) shows that Lookahead can improve convergence of the methods that provably perform well on bilinear games, such as gradient descent with negative momentum and extragradient with momentum.

---

### Official Review · AnonReviewer1 · 2020-10-30
**Require more work in terms of clarity**

**Rating:** 4
**Confidence:** 3

**Review:**

This paper studies the theoretical aspect of lookahead dynamics of smooth games, inspired by the recently introduced Lookahead optimizer, in the spirit of studying the game dynamics of multiple agents.

The overall writing of the paper is not clear enough and should be organized better. For instance, in Sections 3.2 and 4, there are several consecutive theorems which should be better organized to enhance readability. The *Proof.* environments should also be omitted. Not being an expert in this area, I cannot judge the novelty of this work. The usage of the English language should also be improved to avoid grammatical mistakes.


Pros:
- Possibly novel theoretical study of the game dynamics driven by the recently proposed lookahead optimizer.


Cons:
- The main text is not self-contained; see below:
    - Lots of non-standard notation are not defined before being used. And the main text does not contain any pointers to the supplemental material for the list of mathematical notation. The notation $ \mathbb{M}_{m \times m} $ is never defined. How is it different from $ \mathbb{R}^{m \times m} $? Or does it stand for a matrix group like $ \mathrm{GL}_m(\mathbb{R}) $? I also think the principal argument $ \mathrm{Arg} $ should be used instead of vaguely defining $ \mathrm{arg} $ which is multi-valued in complex analysis.
    - Some of the propositions and theorems in the paper are not mathematically precise or rigorous enough. E.g., proper choice of $ k $, small enough $\alpha $, etc. Without making the statements precise, I found some theoretical results in the paper vacuous and hard to interpret.

- The numerical experiments are not sufficient. Results of GAN optimization are expected as in other papers in this line of work, in order to demonstrate the full effectiveness of the proposed scheme.

- The figures, especially Figures 2 and 3, seem to be of quality not up to publication standard and unclear.


Despite my unfamiliarity with this line of work, I think this paper needs to be improved before acceptance, and I suggest rejection for the current state of this paper.

---

> ### Author Response · Authors · 2020-11-19
> **Answers to Reviewer1**
>
> We appreciate the reviewer for constructive comments. Below, we address each concern in detail. Please see blue fonts in the updated version of the manuscript to check how our paper is updated to reflect the comments.
>
> # 1. Organization
> To improve the readability and help readers better understand the actual consequences of our results, in the updated draft, we added paragraphs between the theorems in Section 3.2 and Section 4 to discuss intuitive implications of each result.
>
> # 2. Notations
> We would like to clarify that we summarized our notations in Table A.1 and provided a pointer to the table in introduction of Section 2. In the initial draft, we used \mathbb{M}_{m\times m} for representing the set of real m\times m matrices. However, we agree with the review that some notations, including \mathbb{M}_{m\times m} and \arg, could be misleading. To address Reviewer1’s concerns, we changed some of our notations in the updated version as follows.
>
> - $\mathbb{M}_{m\times m}$ -> $\mathbb{R}^{m\times m}$
> - $\arg$ -> Arg
> - $eig(\textbf{M})$ -> $\lambda(\textbf{M})$
> - $eig_{max}(\textbf{M})$ -> $\lambda_{max}(\textbf{M})$
>
>
> # 3. Non-rigorous terms in theoretical statements
> We acknowledge that the terms such as ‘proper choice of $k$’ and ‘mild assumptions on $k$’ in Proposition 1-2 could be vague. Actually, their detailed descriptions can be found in the proof (Appendix E.1-2). In the newly uploaded draft, we explicitly added more mathematically rigorous terms to Proposition 1-2 as follows.
>
> **Proposition 1.** *… globally converges to the Nash equilibrium if $\Re((1+i\eta)^k) <1$ and $\alpha$ is small enough.*
>
> **Proposition 2.** *… globally converges to the Nash equilibrium of Equation 8 if $\Re((1+i\eta)^k) < (1+\eta^2)^k$ and $\alpha$ is large enough.*
>
> Please note that we leave the expressions like ‘small enough $\alpha$’ as it is quite conventional to use such an expression for stating the fact that ‘there exists $\delta>0$ such that the statements hold for any $\alpha \in (0, \delta)$’ in mathematical arguments.
>
>
> # 4. Insufficient numerical experiments
> 1. We would like to emphasize that the goal of our paper is to establish the theoretical understanding of Lookahead dynamics, rather than proposing a new method to improve GAN training. Extensive large-scale experiments conducted in [1] showed that Lookahead optimizer significantly improves the generative performance of GANs. The goal of this work is to mathematically prove why. It could sound strange, but there is no “our method” in this work, so please refer to [1] for more experimental results.
>
> 2. Given that our main contributions are multiple theorems, the experiments need to focus on their correctness. In this sense, we believe that our experiments are not insufficient. Each experiment verifies our theoretical predictions made by Theorem 5-8 and Theorem 10-11; the Lookahead hyperparameters k and $\alpha$ predicted by our theory indeed stabilize and accelerate the (local) convergence to an equilibrium in both bilinear and nonlinear games.
>
> 3. One relevant “GAN experiment” that we could think of, is to train GANs and see if our assumptions on eigenvalues of Jacobian (i.e., non-zero imaginary parts and imaginary conditioning less than 3) actually hold in practice. To verify this point, we conduct additional experiments and report the results in Appendix F.1 of the updated draft. In Appendix F.1, we train GANs on MNIST dataset and visualize the eigenvalues of gradient descent dynamics. Figure 5 suggests that our assumptions can hold even in a practical nonlinear game like GANs.
>
> # 5. Poor quality of figures
> We updated Figure 2-3 of better qualities and added more description to the captions of Figure 3.
>
> ---
> [1] Tatjana Chavdarova and Matteo Pagliardini and M. Jaggi and François Fleuret. “Taming GANs with Lookahead” arXiv:abs/2006.14567.

---

### Author Response · Authors · 2020-11-24
**Major concerns have been addressed in the updated draft**

We deeply thank the reviewers for constructive and insightful comments.

In the updated draft, we **(1) significantly relax the eigenvalue assumptions** (Lemma 3-4, Theorem 5-8, Theorem 10-11), and **(2) conduct additional experiments** (Appendix F) to show that these assumptions can be realistic even for ill-conditioned bilinear games and GANs. Presentation of our theorems (Theorem 5-8, Theorem 10-11) are greatly improved as well.

Please see blue fonts in the updated version of the paper to see how our paper is updated.

---

### Decision · Program_Chairs · 2021-01-07
**Final Decision**

**Decision:**

Reject

**Comment:**

In this paper, the authors study the behavior of the Lookahead dynamics of Zhang et al. (2019) in bilinear zero-sum games. These dynamics work as follows: given a base algorithm for solving the game (such as gradient descent-ascent or extra-gradient), the Lookahead dynamics perform $k$ iterations of the base algorithm followed by an exponential moving average step with weight $\alpha$. The authors then provide a range of sufficient conditions for the eigenvalues of the matrix defining the game under which the Lookahead dynamics become more stable and converge faster than the base method.

This paper received four reviews and generated a very lively discussion between the authors and reviewers. Reviewer 4 was enthusiastic about the paper; the other three initially recommended rejection. During the discussion phase, the authors revised their paper extensively, and Reviewer 3 increased their score to an "accept" recommendation as a result. In the end, the reviewers were evenly split, and I also struggled a lot to reach a recommendation decision.

On the plus side, the paper treats an interesting problem: prior empirical evidence suggests that the Lookahead dynamics can improve the training of some adversarial machine learning models, so a theoretical study is very welcome and of clear value. On the other hand, the setting treated by the paper (bilinear min-max games) is somewhat restrictive, and the authors' theoretical conclusions do not always admit as clear an interpretation as one would like.

The issues that ended up playing the most important role in my recommendation were as follows:
1. The Lookahead dynamics with period $k$ involve $k$ gradient evaluations, so their rate of convergence should be compared at a $k:1$ ratio to GD and EG (with an additional $2:1$ ratio between GD and EG to put things on an even scale). To a certain degree, this $k:1$ ratio is present in the last part of Lemma 3; however, the exact acceleration achieved by the "shrinkage" of the spectral radius is not clear. This can also be seen in the semi-log plots provided by the authors, where the corresponding slopes of GD/EGD methods should be multiplied by $k$ when compared to the respective LA variants. In this regard, a comparison with the values of $k$ provided in Appendix D reveal that the performance of the Lookahead variants in terms of gradient queries is very similar (if not worse) to the non-LA variants. This is a cause of concern because, if LA does not accelerate convergence in simple bilinear games, it is not credible to expect faster convergence in more complicated problems. During the AC/reviewer discussion of this point, Reviewer 3 pointed out that this might be due to a suboptimal tuning of $\alpha$ (i.e., that it was not chosen "small enough"), and went out to note that this echoes the arguments of other reviewers that the characterization of acceleration may be problematic and not significant (even if it takes place).
2. Another major concern has to do with the stabilization provided by the Lookahead dynamics: using a benchmark game proposed in a recent paper by Hsieh et al. (2020), the authors showed that the Lookahead dynamics converge to a point which is unstable under GDA/EG (and hence avoided). This is fully consistent with the authors' theoretical analysis, but it also highlights an important problem with the Lookahead optimizer: if $k$ and $\alpha$ are tuned to suitable values for stabilization, the algorithm converges to a non-desirable critical point (a max-min instead of a min-max solution). This is a major cause of concern because it shows that the algorithm may, in general, converge to highly suboptimal states.

The above create an inconsistency in the main story of the paper. In fact, it seems to me that the authors' results form more of a "cautionary tale in hiding": even in very simple bilinear problems, the lookahead step may not provide acceleration and, even worse, it could converge to highly undesirable critical points. I find this "negative" contribution quite valuable from a theoretical standpoint, and I believe that a thoroughly revised paper along these lines would be of interest in the top venues of the community (though a more theoretical outlet like COLT might be more appropriate). However, this would require a drastic rewrite of the paper, to the extent that it should be treated as a new submission.

In view of all this, I am recommending a rejection at this stage. I insist however that this should not be seen as a critique for the mathematical analysis of the authors (which was appreciated by the reviewers), but as a recommendation to reframe the paper's narrative to bring it in line with the algorithm's observed behavior. I strongly encourage the authors to resubmit at the next top-tier opportunity.